# How Much Space Has Been Explored? Measuring the Chemical Space Covered by Databases and Machine-Generated Molecules

**Yutong Xie**[1], **Ziqiao Xu**[2], **Jiaqi Ma**[3], **Qiaozhu Mei**[1]*

[1]School of Information, University of Michigan, Ann Arbor, Michigan, USA
[2]Chemistry Department, University of Michigan, Ann Arbor, Michigan, USA
[3]School of Information Sciences, University of Illinois Urbana-Champaign, Champaign, Illinois, USA

## ABSTRACT

Forming a molecular candidate set that contains a wide range of potentially effective compounds is crucial to the success of drug discovery. While most databases and machine-learning-based generation models aim to optimize particular chemical properties, there is limited literature on how to properly measure the coverage of the chemical space by those candidates included or generated. This problem is challenging due to the lack of formal criteria to select good measures of the chemical space. In this paper, we propose a novel evaluation framework for measures of the chemical space based on two analyses: an axiomatic analysis with three intuitive axioms that a good measure should obey, and an empirical analysis on the correlation between a measure and a proxy gold standard. Using this framework, we are able to identify *#Circles*, a new measure of chemical space coverage, which is superior to existing measures both analytically and empirically. We further evaluate how well the existing databases and generation models cover the chemical space in terms of #Circles. The results suggest that many generation models fail to explore a larger space over existing databases, which leads to new opportunities for improving generation models by encouraging exploration.

## 1 INTRODUCTION

To efficiently navigate through the huge chemical space for drug discovery, machine learning (ML) based approaches have been broadly designed and deployed, especially *de novo* molecular generation methods (Elton et al., 2019; Schwalbe-Koda & Gómez-Bombarelli, 2020; Bian & Xie, 2021; Deng et al., 2022). Such generation models learn to generate candidate drug designs by optimizing various *molecular property scores*, such as the binding affinity scores. In practice, these scores can be computationally obtained using biological activity prediction models (Olivecrona et al., 2017; Li et al., 2018), which is the key to obtaining massive labeled training data for machine learning. However, high *in silico* property scores are far from sufficient, as there is usually a considerable misalignment between these scores and the *in vivo* behaviors. Costly wet-lab experiments are still needed to verify potential drug hits, where only a limited number of drug candidates can be tested.

In light of this cost constraint, it is critical to select or generate drug candidates not only with high *in silico* scores, but also covering a large portion of the chemical space. As functional difference between molecules is closely related to their structural difference (Huggins et al., 2011; Wawer et al., 2014), a better coverage of the chemical space will likely lead to a higher chance of hits in wet experiments. For this purpose, quantitative coverage measures of the chemical space become crucial. Such measures can both be used to evaluate and compare the *candidate libraries*[1], and be incorporated into training objectives to encourage ML models better explore the chemical space.

In this paper, we investigate the problem of quantitatively measuring the coverage of the chemical space by a candidate library. There have been a few such coverage measures of chemical space. For example, *richness* counts the number of unique compounds in a molecular set, and it has been used to describe how well a model is able to generate unique structures (Shi & von Itzstein, 2019;

---

*Correspondence to: Qiaozhu Mei <qmei@umich.edu>.

[1]A candidate library is a collection of drug candidates filtered by rules or generated by ML models.

Polykovskiy et al., 2020). In addition, molecular fingerprints have been used to calculate pairwise similarity or distance between two compounds, and the average of these pairwise distances has been used to describe the overall *internal diversity* of a molecular set (Brown et al., 2019; Polykovskiy et al., 2020).

However, most existing coverage measures are heuristically proposed and the validity of these measures is rarely justified. In fact, defining the "right" measure for the coverage of chemical space coverage is challenging. Unlike the molecular property scores, there is no obvious "ground truth" about the coverage of chemical space. Moreover, the chemical space is complex and combinatorial, making the design of a good measure even more difficult.

To address the fundamental problem of properly measuring the coverage of chemical space for drug discovery, we propose a novel evaluation framework with two complementary criteria for evaluating the validity of coverage measures. We first formally define the concept of coverage measures on the chemical space (referred to as *chemical space measures*), where many existing heuristic measures fall into our definition (Section 3). Then we introduce the two criteria and compare various (existing and new) chemical space measures based on the criteria (Section 4). Specifically, the first criterion (Section 4.1) is based on an axiomatic analysis with three intuitive axioms that a good chemical space measure should satisfy. Surprisingly, most heuristic measures that are commonly used in literature, such as *internal diversity*, fail to satisfy these intuitive axioms. The second criterion (Section 4.2) compares the chemical space measures with a proxy of the gold standard: the number of unique biological functionalities covered by the set of molecules. We find that #Circles, a new chemical space coverage measure (defined in Section 3.2.3) that has a strong basis in the mathematical literature, not only satisfies both axioms but also better correlates with the gold standard.

Finally, we apply the #Circles measure to evaluate how well the existing databases and ML models cover the chemical space (Section 5). Interestingly, the evaluation results suggest that many ML models fail to explore a larger portion of chemical space compared to drug candidates obtained from virtual screening over existing databases. We believe these findings lead to a new direction to improve ML-based drug candidate generation models on better exploring the chemical space.

## 2 RELATED WORK

Molecular databases and machine-generated compounds are rich sources of drug candidates for forming a candidate library in drug discovery. To evaluate the quality of molecular databases and molecular generation methods, a variety of metrics are proposed. In general, four categories of evaluation metrics can be identified in the literature, which are related to: (1) bioactivities, (2) molecular properties, (3) data likelihood, or (4) the coverage of the chemical space, respectively.

In this paper, we mainly focus on the fourth category of metrics, the metrics that are more or less related to the degree of coverage (or exploration) in the chemical space (other metrics are discussed in Appendix A). In this category, commonly used measures include richness, uniqueness, internal diversity, external diversity, KL divergence, and Fréchet ChemNet Distance (FCD) (Olivecrona et al., 2017; You et al., 2018; De Cao & Kipf, 2018; Elton et al., 2019; Brown et al., 2019; Popova et al., 2019; Polykovskiy et al., 2020; Shi et al., 2020; Jin et al., 2020; Xie et al., 2021). Besides, Zhang et al. (2021) propose to use the number of unique functional groups or ring systems to estimate the chemical space coverage and to compare several recent generative models. Similarly in Blaschke et al. (2020), the number of unique Bemis-Murcko scaffolds is used to measure the variety of drug candidates. Koutsoukas et al. (2014) study the effect of molecular fingerprinting schemes on the internal diversity of compound selection. These measures usually mix the concepts of diversity, coverage, or novelty, and their validity as a measure of exploration is not justified.

To the best of our knowledge, this is the first work that formally investigates the validity of molecular chemical space measures. In particular, axiomatic approaches are used to analytically evaluate various designs of a measurement, such as utility functions (Herstein & Milnor, 1953), cohesiveness (Alcalde-Unzu & Vorsatz, 2013), or document relevance (Fang et al., 2004). While one study applies axiomatic analysis to the design of diversity measures, with a particular focus on the domain of science of science (Yan, 2021), the analysis of chemical space measurements remains novel. Using axiomatic analysis to evaluate the chemical space measures in the chemical space is novel. With an empirical analysis in addition to the axiomatic analysis, we make practical recommendations on effective chemical space measures, including two novel measures.

## 3  DEFINING CHEMICAL SPACE MEASURES

### 3.1  DEFINITION OF CHEMICAL SPACE MEASURES

To define a chemical space measure, we first formalize the notion of chemical space by assuming a distance metric exists. This assumption is widely adopted in cheminformatics, using distance metrics such as the Tanimoto distance (Tanimoto, 1968; Bajusz et al., 2015).

**Assumption 3.1** (Chemical space). The chemical space $\mathcal{U}$ contains all possible molecules and is a metric space with a distance metric function $d : \mathcal{U} \times \mathcal{U} \to [0, +\infty)$.

**Definition 3.2** (Tanimoto distance). For two molecules $x_1, x_2 \in \mathcal{U}$, whose binary molecular fingerprint vectors are $\boldsymbol{x}_1, \boldsymbol{x}_2 \in \{0,1\}^n$ where $n$ is the dimensionality of the fingerprint, their Tanimoto distance is defined as

$$d(x_1, x_2) := 1 - \frac{\sum_{j=1}^n \boldsymbol{x}_{1j} \cdot \boldsymbol{x}_{2j}}{\sum_{j=1}^n \max(\boldsymbol{x}_{1j}, \boldsymbol{x}_{2j})}. \tag{1}$$

The Tanimoto distance is also referred to as the Jaccard distance (Jaccard, 1912) in other domains. Its range is $[0, 1]$. For finite sets (*e.g.*, molecular fingerprints), the Tanimoto distance is a metric function (Kosub, 2019; Lipkus, 1999).

Our definition and following analysis are generic to all distance functions and are not limited to the Tanimoto distance. One can also use the distance of latent hidden vectors (Preuer et al., 2018; Samanta et al., 2020) or the root-mean-square deviation (RMSD) of three-dimensional molecular conformers (Fukutani et al., 2021).

We then define a chemical space measure as the following.

**Definition 3.3** (chemical space measure). Given the universal chemical space $\mathcal{U}$, a chemical space measure is a function that maps a set of molecules to a non-negative real number that reflects to what extent the set spans the chemical space, *i.e.*, $\mu : \mathcal{P}(\mathcal{U}) \to [0, \infty)$, where $\mathcal{P}(\cdot)$ is the notation of power set. In particular, $\mu(\emptyset) = 0$.

### 3.2  EXAMPLES OF CHEMICAL SPACE MEASURES

Definition 3.3 is intentionally left general. Many measures related to the coverage, diversity, or novelty of a molecular set fall into this definition. We summarize these various measures used in literature into three categories: *reference-based* measures, *aggregation-based* measures, and *locality-based* measures. We are also able to define new chemical space measures under this formulation.

#### 3.2.1  REFERENCE-BASED MEASURES

The first broad category of chemical space measures compare the generated molecules $\mathcal{S}$ with a reference set $\mathcal{R}$. In such context, a reference-based measure can be defined as the coverage of references:

$$\text{Coverage}(\mathcal{S}, \mathcal{R}) := \sum_{y \in \mathcal{R}} \left( \max_{x \in \mathcal{S}} \text{ cover}(x, y) \right), \tag{2}$$

where $\text{cover}(x, y)$ indicates how molecule $x$ can cover the reference $y$.

When the reference set $\mathcal{R}$ is taken as a collection of molecular fragments, the coverage function can be written as $\text{cover}(x, y) := \mathbb{I}[\text{molecule } x \text{ contains fragment } y]$, where $\mathbb{I}[\cdot]$ is the indication function. A large body of drug discovery literature uses the number of distinct functional groups (FG), ring systems (RS), or Bemis-Murcko scaffolds (BM) in $\mathcal{S}$ to gauge the size of explored chemical space (Zhang et al., 2021; Blaschke et al., 2020), corresponding to the cases where $\mathcal{R}$ is the collection of all possible FG, RS, or BM fragments. We denote these specific reference-based measures as #FG, #RS, and #BM respectively. Another example is the Richness of the molecular set, *i.e.*, Richness $:= |\mathcal{S}|$ (Shi & von Itzstein, 2019; Polykovskiy et al., 2020), where $\text{cover}(x, y) := \mathbb{I}[x = y]$ and $\mathcal{R} = \mathcal{U}$.

#### 3.2.2  AGGREGATION-BASED MEASURES

Though *reference-based* measures estimate chemical space coverage in an intuitive way, such measures highly rely on the characteristics of the reference set and do not consider the relations between the compounds. In contrast, the *aggregation-based* measures utilize the pair-wise distances

among molecules and reflect the extent of chemical space coverage by aggregating the distances. For a set $\mathcal{S}$ with $n$ molecules, several *aggregation-based* measures can be defined as follow:

$$\text{Diversity}(\mathcal{S}; d) := \frac{2}{n(n-1)} \sum_{\substack{x,y \in \mathcal{S} \\ x \neq y}} d(x,y), \quad \text{(3a)} \qquad \text{SumDiversity}(\mathcal{S}; d) := \frac{1}{n-1} \sum_{\substack{x,y \in \mathcal{S} \\ y \neq x}} d(x,y), \quad \text{(3d)}$$

$$\text{Diameter}(\mathcal{S}; d) := \max_{\substack{x,y \in \mathcal{S} \\ x \neq y}} d(x,y), \qquad \text{(3b)} \qquad \text{SumDiameter}(\mathcal{S}; d) := \sum_{x \in \mathcal{S}} \max_{\substack{y \in \mathcal{S} \\ y \neq x}} d(x,y), \qquad \text{(3e)}$$

$$\text{Bottleneck}(\mathcal{S}; d) := \min_{\substack{x,y \in \mathcal{S} \\ x \neq y}} d(x,y), \qquad \text{(3c)} \qquad \text{SumBottleneck}(\mathcal{S}; d) := \sum_{x \in \mathcal{S}} \min_{\substack{y \in \mathcal{S} \\ y \neq x}} d(x,y), \qquad \text{(3f)}$$

where $x, y$ are molecules in $\mathcal{S}$, $d$ is a distance metric as defined previously.

Among these measures, Diversity (also referred to as the *internal diversity*), the average distance between the molecules, is widely used in the literature (You et al., 2018; De Cao & Kipf, 2018; Popova et al., 2019; Polykovskiy et al., 2020; Shi et al., 2020; Xie et al., 2021). We follow the topology theory and also introduce Diameter and Bottleneck as the maximum and minimum distance between any pair of molecules, respectively (Edelsbrunner & Harer, 2010), since they can also reflect the dissimilarity between molecules in $\mathcal{S}$. We further introduce three Sum- variants for the above measures. The Sum- variants[2] will tend to increase when new molecules are added into the set. In addition, the determinant of the similarity matrix of molecules is also a measure of dissimilarity, which is often employed in diverse subset selection as a key concept of the determinantal point processes (DPP) (Kulesza & Taskar, 2011; Kulesza et al., 2012). The DPP measures is defined as $\text{DPP}(\mathcal{S}) := \det(\boldsymbol{S})$, where $\boldsymbol{S}$ is the Tanimoto similarity matrix of candidate molecules (*i.e.*, $1 - d(x,y)$ for Tanimoto distance).

### 3.2.3 LOCALITY-BASED MEASURES

Inspired by the sphere exclusion algorithm used in compound selection (Snarey et al., 1997; Gobbi & Lee, 2003), we introduce a new chemical space coverage measure that highlights the local neighborhoods covered by a set of molecules:

$$\#\text{Circles}(\mathcal{S}; d, t) := \max_{\mathcal{C} \subseteq \mathcal{S}} |\mathcal{C}| \quad \text{s.t.} \quad d(x,y) > t, \quad \forall x \neq y \in \mathcal{C}, \tag{4}$$

where $t \in [0, 1)$ is a distance threshold that corresponds to the diameter of a circle.

Intuitively, #Circles counts the maximum number of mutually exclusive circles that can fit into $\mathcal{S}$ as neighborhoods, with a subset of its members $\mathcal{C}$ as the circle centers. When the threshold $t = 0$, #Circles becomes the richness measure $\text{Richness}(\mathcal{S}) := |\mathcal{S}|$, which is the number of unique molecules (Shi & von Itzstein, 2019; Polykovskiy et al., 2020). Interestingly, #Circles has a close relation to the concepts of *covering* and *packing* in mathematics, which is in particular related to the definition of *packing number* in topology (Vershynin, 2018). We discuss their detailed connections in Appendix B.

## 4 SELECTING THE RIGHT MEASURE

While all the aforementioned measures can heuristically reflect the degree of exploration, they do not always agree with each other. We need a principled way to select the most suitable measures from a variety of possible choices. Ideally, the selection criteria should not depend on the particular molecule properties of the target or the algorithm used to generate the candidates. We achieve this through a qualitative axiomatic analysis and a quantitative correlation comparison.

### 4.1 CRITERION #1: AN AXIOMATIC ANALYSIS OF CHEMICAL SPACE MEASURES

We propose three simple and intuitive principles that a good chemical space measure should satisfy. First, including more molecules should not decrease the degree of chemical space coverage. As it is easy to filter large candidate libraries into smaller subsets, including more molecules (even similar ones) is not harmful, and can render a higher probability of containing drug hits. Second, when adding new molecules, the coverage should increase properly, instead of inflating too much. Third, molecular sets with more dissimilar molecules should have a higher degree of chemical space coverage. These three principles are formalized below as three axioms and can be tested analytically.

---

[2]The SumBottleneck measure is sometimes abbreviated as "SumBot" in the main paper.

**Axiom 4.1** (Monotonicity). *A good chemical space measure $\mu$ should be monotonic, i.e., for any two molecular sets $\mathcal{S}_1, \mathcal{S}_2 \subseteq \mathcal{U}$, it holds that*

$$\mu(\mathcal{S}_1 \cup \mathcal{S}_2) \geq \max(\mu(\mathcal{S}_1), \mu(\mathcal{S}_2)). \tag{5}$$

A monotonic measure tends to increase when more molecules are included. This tendency is intuitive in drug discovery as testing more molecules means having a higher probability to discover a drug hit.

**Axiom 4.2** (Subadditivity). *A good chemical space measure $\mu$ should be subadditive, i.e., for any two molecular sets $\mathcal{S}_1, \mathcal{S}_2 \subseteq \mathcal{U}$, it holds that*

$$\mu(\mathcal{S}_1 \cup \mathcal{S}_2) \leq \mu(\mathcal{S}_1) + \mu(\mathcal{S}_2). \tag{6}$$

Note that the coverage of chemical space spanned by a molecular set may correlate with the probability of containing drug hits. So correspondingly, the coverage $\mu(\mathcal{S}_1 \cup \mathcal{S}_2)$ should not exceed the sum of $\mu(\mathcal{S}_1)$ and $\mu(\mathcal{S}_2)$. Some direct corollaries and discussions on how the first and second axioms connect to mathematical measures are in Appendix C.

**Axiom 4.3** (Dissimilarity). *A good chemical space measure should have a preference to dissimilar elements, i.e., for any three molecules $x_0, x_1, x_2 \in \mathcal{U}$, if $d(x_0, x_1) \geq d(x_0, x_2)$, it holds*

$$\mu(\{x_0, x_1\}) \geq \mu(\{x_0, x_2\}), \tag{7}$$

*where $d$ is the distance metric.*

Intuitively, considering adding a new molecule into an existing molecular set $\mathcal{S} = \{x_0\}$, where there are two choices, namely $x_1$ and $x_2$, the more dissimilar one $x_1$ will be preferred.

Depending on whether each chemical space measure satisfies *monotocinicy*, *subadditivity* and/or *dissimilarity*, we can put it into a Venn diagram as shown in Figure 1. The formal proofs are provided in Appendix D. Despite that the three axioms are simple and intuitive, surprisingly, the proposed #Circles is the only measure that satisfies all these three axioms.

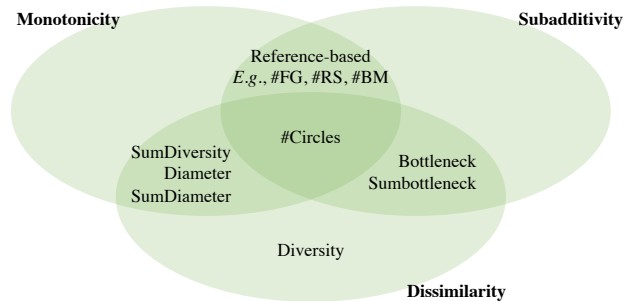

Figure 1: Chemical space measures, categorized by whether they satisfy monotonicity, subadditivity, and/or dissimilarity. #Circles is the only measure that satisfies all simultaneously.

## 4.2 CRITERION #2: CORRELATION WITH BIOLOGICAL FUNCTIONALITY

While only one measure stands out in the axiomatic analysis, it does not guarantee its empirical performance. Meanwhile, there are still multiple measures that satisfy *one or two* of the three axioms.

In this subsection, we further investigate the validity of the chemical space measures by testing their correlations with the biological functionality variety of molecules. Such functionalities can provide valuable information in distinguishing molecules and implicating chemical space coverage. We correlate the chemical space measures to the number of unique biological functionality labels covered by a molecular set. A better chemical space measure should have a higher correlation to the variety of biological functionalities, even though it is a proxy gold standard.

### 4.2.1 EXPERIMENT SETUP

We base the analysis on the BioActivity dataset that is also used to compare different compound selection algorithms (Koutsoukas et al., 2014). This dataset contains 10,000 compound samples extracted from the ChEMBL database (Gaulton et al., 2017) with bio-activity labels, which annotate 50 activity classes with 200 samples each. Following Koutsoukas et al. (2014), for a subset of this dataset $\mathcal{S}$, we take the number of unique class labels as a proxy "gold standard" of the variety of the molecules in $\mathcal{S}$, *i.e.*, $\text{GS}(\mathcal{S}) := \#$unique labels in $\mathcal{S}$, which represents the number of biological functionality types covered by $\mathcal{S}$. We then compare the behavior of the gold standard and the chemical space measures in two settings to find out which measures have the highest correlations with the coverage (or variety) of biological functionalities: (1) a *fixed-size* setting where random subsets of the dataset with fixed sizes are measured, and (2) a *growing-size* setting where we sequentially

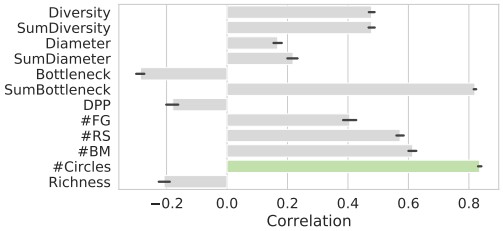 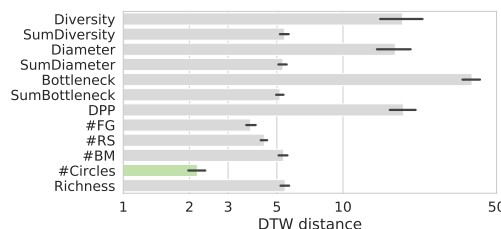

(a) Spearman's correlations between the gold standard GS and chemical space measures in the fixed-size setting. A higher correlation is better.

(b) DTW distances between the gold standard GS and chemical space measures in the growing-size setting, the smaller the better. A smaller distance is better.

Figure 2: Correlation between each chemical space measure and biological functionality coverage. The chemical space measure with the highest correlation is highlighted in green. Results are aggregated by running experiments independently ten times.

add molecules into a molecular set. We compare the measuring results of GS and chemical space measures with Spearman's correlation coefficient and dynamic time warping (DTW) distance. The experiment details are listed in Appendix E.

#### 4.2.2 RESULTS AND DISCUSSION

Experiment results in Figure 2 suggest that #Circles stands out for both empirical experiment settings. For the *fixed-size* setting, #Circles and SumBottleneck have notably higher correlations to the gold standard than other measures. This indicates that the locality information is critical, as these two measures both prefer new molecules that are at arm's length from their nearest neighbors. For the *growing-size* setting, #Circles surpasses all other chemical space measures.

Table 1: Regarding the two suggested criteria, **#Circles** is the most recommended chemical space measure. "Mono", "subadd", and "dissim" are abbreviations for monotonicity, subadditivity, and dissimilarity respectively.

| | *(C1) Axiomatic properties (Sec. 4.1)* | *(C2) Correla -tions (Sec. 4.2)* | |
|---|---|---|---|
| **Measures** | *Axioms satisfied* | *Fixed* | *Growing* |
| Diversity | Dissimilarity | Medium | Low |
| SumBot | Subadd, dissim | High | Medium |
| #FG | Mono, subadd | Medium | Medium |
| Richness | Mono, subadd, dissim | Low | Medium |
| **#Circles** | Mono, subadd, dissim | High | High |

Reference-based measures such as #FG also perform prominently. Both #Circles and reference-based measures satisfy subadditivity, making them suitable for a "growth" setting. The Sum- variants of aggregation-based measures outperform their original forms, as they tend to increase when adding new molecules.

More experiment results can be found in Appendix E, where we also discuss the impact of the distance metric $d$. We also study the sensitivity of the subset size $n$, the way of adding new molecules, and the choice of the #Circles's threshold $t$. Results suggest that $t = 0.75$ is a good choice for both two settings. Correlations between chemical space measures are visualized in Figure 8 and 11.

### 4.3 WHICH MEASURE IS THE RIGHT CHOICE?

The empirical analysis shows that the locality-based #Circles measure is a robust choice for all tested scenarios, which reconfirms the conclusion of the axiomatic analysis. Besides, SumBottleneck may be an effective choice when a fixed number of candidates are the target. Reference-based measures may be good alternatives if carefully-designed and comprehensive reference sets are available. Surprisingly, the widely used Diversity measure is rendered inferior both analytically and empirically, casting doubts on its efficiency in measuring and encouraging exploration for molecular generation. Our analysis and comparison results are summarized in Table 1.

## 5 MEASURING CHEMICAL SPACE COVERAGE

In this section, we apply the chemical space measures to evaluate how well the existing databases and ML-based molecular generation models cover the chemical space.

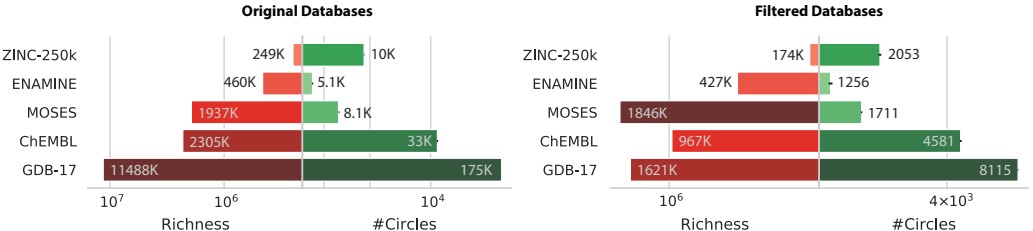

Figure 3: Richness and #Circles of molecular databases. Databases are listed vertically. The horizontal axes are chemical space measures (plotted in logarithmic scales). #Circles has the threshold value ($t = 0.75$). We find a discrepancy between Richness and #Circles, meaning that a larger database does not necessarily span a wider range of the chemical space.

## 5.1 Measuring Chemical Space Covered by Molecular Databases

Other than being a rich source for virtual screening for drug purposes, molecular databases are also bonanzas of data for training ML models. In both scenarios, the coverage of chemical space is a crucial descriptor for comparing various databases. With a higher coverage of the chemical space, we are able to construct candidate libraries with better varieties. Also, the ML model would benefit from the diverse training data, obtaining a better understanding of the overall chemical space. Here, we compare five commonly used molecular databases with the selected measures.

### 5.1.1 Experiment Setup

**Databases and filtering rules.** We include five widely used drug and compound databases in our measurement: (1) **ZINC-250k** (a random subset of the ZINC database) (Irwin & Shoichet, 2005), (2) **MOSES** (Polykovskiy et al., 2020), (3) **ChEMBL** (Gaulton et al., 2017), (4) **GDB-17** (Ruddigkeit et al., 2012), and (5) the **Enamine Hit Locator Library** (ENAMINE). More details about the databases are listed in Appendix F.1.

We further apply two filtering rules to these databases to select potential drug candidates. In drug discovery, drug-likeness and synthesizability of compounds are both important considerations. The quantitative drug-likeness (QED) (Bickerton et al., 2012) and the synthetic accessibility (SA) (Ertl & Schuffenhauer, 2009) are computed. The molecules are filtered with QED $\geq 0.6$ and SA $\leq 4$.

### 5.1.2 Results and Discussion

Figure 3 compares Richness and Circles of molecular databases. From the results, we conclude the findings as follows. (1) **A larger database does not necessarily span a wider range of the chemical space**: There is a discrepancy between Richness and #Circles, and the #Circles measuring results suggest that, in terms of chemical space coverage, ZINC-250k covers a larger area than ENAMINE and MOSES. (2) **ZINC-250k and GDB-17 are recommended**: With the smallest amount of compounds, ZINC-250k renders a relatively high coverage of the chemical space, which is highly recommended for model training with limited computation resources. Constructed by enumeration, GDB-17 substantially expands the known chemical space to a great level, which is suitable for training ML models for chemical space representations. More details and findings on other chemical measures are provided in Appendix F.

## 5.2 Measuring Chemical Space Explored by Generation Models

### 5.2.1 Problem Formulation

In drug discovery, the search for active molecules towards a druggable target is often formulated as the following unconstrained optimization problem (Olivecrona et al., 2017; Gómez-Bombarelli et al., 2018; Liu et al., 2018; You et al., 2018; Jin et al., 2018; De Cao & Kipf, 2018; Popova et al., 2019; Shi et al., 2020; Xie et al., 2021):

$$\underset{x \in \mathcal{U}}{\arg \max} \quad s_1(x) \circ s_2(x) \circ \cdots s_K(x), \tag{8}$$

where $x$ is a molecule in the chemical space $\mathcal{U}$, $s_k : \mathcal{U} \to \mathbb{R}$ is a function scoring particular biological properties, and the "$\circ$" operator indicates the combination of multiple scores (*e.g.*, summation or multiplication). This scoring function can be binding affinity to protein targets, drug-likeness, synthesizability, *etc*. When wet-lab experiments are not available, these scores are obtained computationally.

Table 2: Measuring results of the chemical space explored by molecular generation methods. Incorporating chemical space measures into the objectives increases the exploration of the chemical space measures, but none of the tested ML-based models surpasses the virtual screening baseline (**Databases**) in terms of **#Circles** when $t = 0.75$ (as suggested in Sec. 4.2, highlighted in grey). Numbers in the parentheses following **#Circles** are values for the threshold $t$. In each chemical space measure, the larger value the better. **Bold** indicates the best performance in each measure.

| Method | Richness | #Circles (0.70) | #Circles (0.75) | #Circles (0.80) | Diversity |
|---|---|---|---|---|---|
| **Databases** | 1,250 | 78.0 $\pm$ 4.4% | **55.7** $\pm$ 3.7% | **30.3** $\pm$ 5.0% | **0.827** |
| **RationaleRL** | 526 $\pm$ 1.7% | 14.2 $\pm$ 6.8% | 12.1 $\pm$ 6.5% | 9.9 $\pm$ 0.3% | 0.762 $\pm$ 0.3% |
| **DST** | 29 | 2.0 $\pm$ 0.0% | 1.0 $\pm$ 0.0% | 1.0 $\pm$ 0.0% | 0.559 |
| **JANUS** | 261 | 15.0 $\pm$ 11.6% | 6.7 $\pm$ 8.7% | 4.0 $\pm$ 25.0% | 0.759 |
| **MARS** | 134K $\pm$ 10.2% | 75.3 $\pm$ 4.4% | 22.6 $\pm$ 13.3% | 7.3 $\pm$ 16.0% | 0.736 $\pm$ 0.8% |
| +Diversity | 177K $\pm$ 13.7% | 93.8 $\pm$ 7.6% | 24.4 $\pm$ 9.8% | 8.8 $\pm$ 13.9% | 0.752 $\pm$ 0.2% |
| +SumBot | **221K** $\pm$ 13.4% | 128.3 $\pm$ 8.1% | 29.3 $\pm$ 7.6% | 8.8 $\pm$ 13.0% | 0.752 $\pm$ 0.3% |
| +#Circles | 179K $\pm$ 17.4% | **129.9** $\pm$ 19.1% | 30.2 $\pm$ 18.0% | 9.7 $\pm$ 16.2% | 0.749 $\pm$ 0.7% |

Unfortunately, the majority of the computational scoring functions are merely approximations and cannot accurately predict the wet-lab results. As pointed out in the chemistry literature, rather than focusing on optimizing the estimated property scores, it is crucial to generate a variety of compounds that span a wider range of the chemical space (Huggins et al., 2011; Wawer et al., 2014; Ashenden, 2018). We therefore propose the following objective as the goal of molecular generation, and use this new objective to compare models:

$$\arg\max_{\mathcal{S} \subseteq \mathcal{U}} \quad \mu(\mathcal{S}) \quad \text{s.t.} \quad s_k(x) \geq C_k, \ \forall k \in [K], \ x \in \mathcal{S}, \tag{9}$$

where $\mathcal{S} \subseteq \mathcal{U}$ is a molecular candidate set, $\mu(\cdot)$ is a chemical space measure, $s_k(x) \geq C_k$ means the estimated property score of a molecule $x$ is at or above a threshold $C_k$.

### 5.2.2 EXPERIMENT SETUP

**Generation objective.** Following Li et al. (2018) and Jin et al. (2020), we consider the inhibition against an Alzheimer-related target protein c-Jun N-terminal kinase-3 (JNK3) as the biological objective. The JNK3 binding affinity score is predicted by a random forest model based on Morgan fingerprint features of a molecule (Rogers & Hahn, 2010). We also consider drug likeness (QED) and synthetic accessibility (SA) as they are important in practical drug discovery scenarios.

**Models, variations, and the baseline.** We employ four recently proposed molecular generation models to optimize the generation objective (JNK3, QED, and SA): (1) **RationaleRL** (Jin et al., 2020) generates molecules by combining rationales with reinforcement learning (RL); (2) **DST** (Fu et al., 2022) supports a gradient-based optimization on chemical graphs; (3) **JANUS** (Nigam et al., 2022) uses the genetic algorithm (GA) to design drugs inversely with parallel tempering; (4) **MARS** (Xie et al., 2021) generates fragment-based molecular structures based on Markov chain Monte Carlo (MCMC) sampling. Due to MARS' great ability to simultaneously optimize multiple drug discovery objectives, we also consider three variants of it (MARS+Diversity, MARS+SumBottleneck, and MARS+#Circles), where the chemical space measures are incorporated into Equation 8 to encourage exploration. The virtual screening setting is introduced as a baseline, where the five aforementioned **molecular databases** are combined, forming a candidate set of 11M compounds.

**Evaluation.** To validate the ability in exploring the chemical space, we compare the molecules generated by models using Equation 9 with multiple chemical space measures. The molecules (including molecules generated by models as well as compounds from databases) are filtered with the constraints JNK3 $\geq 0.5$, QED $\geq 0.6$, and SA $\leq 4$, representing the positive candidates. We examine seven different measures, and report #Circles, Richness, and Diversity here; the others are listed in Appendix G). Note we cannot directly measure the biological functionality coverage as in Section 4.2, as many of the generated molecules are not in the database. For RationaleRL and DST, 5000 molecules are generated as suggested in their papers. For JANUS and MARS, the population size and the number of MCMC chains are set as 5000. JANUS evolves for 10 generations, and MARS iterates 2000 steps. More implementation details are listed in Appendix G.

### 5.2.3 RESULTS AND DISCUSSION

Table 2 lists the chemical space measures estimated for the positive candidate compounds (*i.e.*, JNK3 $\geq$ 0.5, QED $\geq$ 0.6, and SA $\leq$ 4) generated by models or from databases. Based on these results, we conclude the following. (1) **ML-based models fail to explore a larger effectual area compared to databases**: Unexpectedly, we find the molecules obtained by virtual screening over databases span the largest area of the chemical space in terms of #Circles when $t = 0.75$. The most competitive model among the tested ones, MARS+#Circles, explores only about half of the space compared to virtual screening. (2) **Chemical space measures can encourage the model to explore**: By incorporating chemical space measures into the optimization objective of MARS, all measures are significantly improved during molecular generation. (3) **Diver-**

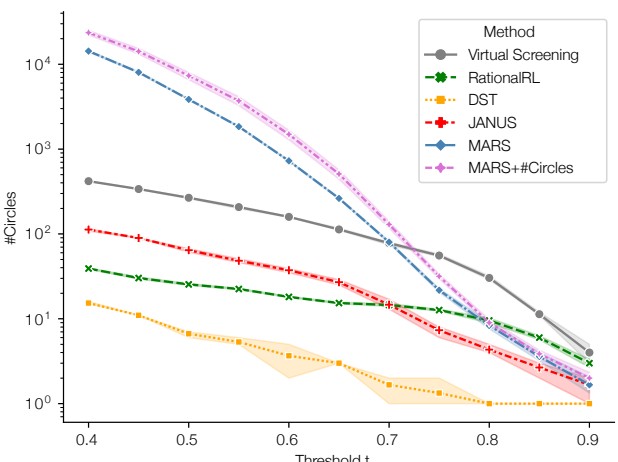

Figure 4: #Circles in the log scale and its threshold $t$ selection for various models. Curves of the #Circles measure reflects characteristics of molecular generation methods.

**sity should be avoided as a descriptor for exploration**: During sampling, MARS and its variants all converge quickly in Diversity (Figure 19), despite a large number of new molecules discovered. Again, it suggests that the widely used Diversity is not suitable for measuring chemical space.

We want to stress that **the parameter $t$ of #Circles has an interpretable meaning**. When $t$ is smaller, #Circles tends to reflect the "spread" of molecules at a finer granularity. When $t$ is larger, the measure focuses more on the global picture. Therefore, we can intuitively characterize different molecular generation methods by examining #Circles values with different thresholds $t$ as displayed in Figure 4. From the results, we find MARS and JANUS present similar characteristics as they both have a relatively sharp decrease around $t = 0.75$. This might be because both two models are based on sampling algorithms and tend to *exploit locally* in the chemical space. In contrast, RationaleRL tends to *explore globally* because it is designed to combine active fragments (rationales). We can also compare the models by adopting the concept of *Pareto optimization*. We find the DST, JANUS, and the vanilla MARS model are completely dominated by the variant MARS+#Circles. Besides, the RationaleRL method is dominated by the virtual screening approach.

A few limitations of the measurement with #Circles should be noted. For example the running time of calculating #Circles is exponential in theory. We implement a fast approximation by sacrificing accuracy (Appendix H). More experiment details and findings are provided in Appendix G.

## 6 CONCLUSION AND FUTURE WORK

In this paper, we have presented a systematic study on the coverage measures of chemical space in the context of drug discovery. The core contribution of this study is a novel evaluation framework for selecting proper coverage measures. Using the proposed framework, we have identified a new chemical space coverage measure, #Circles, which is superior to existing heuristic measures commonly used in the drug discovery literature. We have also quantitatively compared molecular databases and ML-based generation models in terms of #Circles. The results suggest that many ML-based generation models fail to explore a larger chemical space compared to virtual screening, since they tend to exploit locally instead of exploring widely.

Measuring the chemical space is a fundamental problem in drug discovery. Our work in this direction opens even more research questions that are worth investigating in the future. For example, it would be interesting to design and evaluate more chemical space measures under our framework. Moreover, the newly suggested objective for molecular generation (Equation 9) inspires further study, since it is computationally challenging as a combinatorial constrained optimization problem. In general, the proposed chemical space measures and the generic measure selection framework are not restricted to the application of drug discovery, but can also be used in other domains, such as material design, science of science, and text or image generation.

ACKNOWLEDGMENTS

We would like to thank Professor Kevyn Collins-Thompson, Professor Paramveer Dhillon, and Professor Aaron Frank for their helpful feedback and suggestions. We also thank the anonymous reviewers for their constructive comments, and in particular for pointing out the relation between #Circles and packing number. This work was in part supported by the National Science Foundation under grant number 1633370.

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

# A    EVALUATION METRICS FOR MOLECULAR DATABASES AND MOLECULAR GENERATION METHODS

Molecular databases and machine-generated compounds are rich sources of drug candidates for forming a candidate library in drug discovery. To evaluate the quality of molecular databases and molecular generation methods, a variety of metrics are proposed. In general, four categories of evaluation metrics can be identified in the literature, which are related to: (1) bioactivities, (2) molecular properties, (3) data likelihood, or (4) the coverage of the chemical space, respectively.

The first category describes the biological activities of the included or generated molecules toward certain protein targets. Using docking simulators, researchers can calculate docking scores to determine the binding affinity between the given compound structure and the target (Trott & Olson, 2010). However, the simulation usually takes a long time to run. To obtain fast feedback and a sufficient amount of data to train molecular generation models, machine learning (ML) methods are also proposed to predict the bioactivity of molecules, including linear regression, supported vector machines (SVMs), random forests, and neural networks (NNs) (Shoichet et al., 1992; Olivecrona et al., 2017; Li et al., 2018; Ahmadi et al., 2021; Jiang et al., 2022). The second category of metrics evaluates the molecular properties of compounds. Some of them are calculated based on heuristic rules, such as the validity of compounds, molecular weight, octanol-water partition coefficient (logP) (Wildman & Crippen, 1999), drug-likeness score (QED) (Bickerton et al., 2012), and synthetic accessibility (SA) (Ertl & Schuffenhauer, 2009). With the advent of ML and deep learning, ML (especially deep learning)-based prediction models like SVMs, random forests, recurrent neural networks (RNNs), and graph neural networks (GNNs) are also employed to predict the properties of compounds (Shen et al., 2010; Wang et al., 2019; Gilmer et al., 2017). Another branch of metrics uses data likelihood to evaluate the molecules output by generative models (Gómez-Bombarelli et al., 2018; Liu et al., 2018; Jin et al., 2018; De Cao & Kipf, 2018). A few other metrics are particularly related to the coverage of the chemical space or the extent of exploration in the chemical space. These coverage-and-exploration-related metrics are introduced in Section 2.

# B    COVERING NUMBER AND PACKING NUMBER

In mathematics, given a metric space $(T, d)$, where $T$ is the universal set and $d$ is the distance metric, the coverage of a subset $K \subset T$ can be described in terms of $\varepsilon$-*nets* (Vershynin, 2018) defined below.

**Definition B.1** ($\varepsilon$-net). Given a metric space $(T, d)$, for a subset $K \subseteq T$ and a positive number $\varepsilon > 0$, an $\varepsilon$-*net* of $K$ is a subset $\mathcal{N} \subseteq K$, where every point in $K$ is within distance $\varepsilon$ of $\mathcal{N}$, *i.e.*,

$$\forall x \in K, \exists x_0 \in \mathcal{N} : \ d(x, x_0) \leq \varepsilon.$$

Two related concepts, *covering number* and the *packing number*, are then defined as follows.

**Definition B.2** (Covering number). For a subset $K$ of the metric space $(T, d)$, its *covering number*, denoted $\mathcal{N}(K, d, \varepsilon)$, is the smallest possible cardinality of an $\varepsilon$-net of $K$.

**Definition B.3** (Packing number). For a subset $\mathcal{N}$ in the metric space $(T, d)$, it is called $\varepsilon$-*separated* if $d(x, y) > \varepsilon$ for all points $x \neq y \in N$. The *packing number* of a subset $K \subseteq T$, denoted $\mathcal{P}(K, d, \varepsilon)$, is the largest possible cardinality of an $\varepsilon$-separated subset of $K$.

Our definition of #Circles is equivalent to the packing number when setting the distance threshold $t$ in #Circles equal to the diameter of the balls $\varepsilon$ in the packing number. To our best knowledge, however, the use of this notion as a chemical space coverage measure is novel. In light of this connection, further leveraging insights from topology to investigate the chemical space would be an interesting future direction.

# C    AXIOM COROLLARIES AND DISCUSSIONS

Some direct corollaries of monotonicity and subadditivity are as follows.

**Corollary C.1** (Subtraction). *If a chemical space measure $\mu$ is subadditive, then for any two molecular sets $\forall \mathcal{S}_1, \mathcal{S}_2 \subseteq \mathcal{U}$, we have*

$$\mu(\mathcal{S}_1) \geq \mu(\mathcal{S}_1 \setminus \mathcal{S}_2) \geq \mu(\mathcal{S}_1) - \mu(\mathcal{S}_2).$$

**Corollary C.2** (Monotonicity [Single Molecule]). *] If a chemical space measure $\mu$ is subadditive, then for any molecular set $\forall \mathcal{S} \subseteq \mathcal{U}$ and a single molecule $x \in \mathcal{U}$, we have $\mu(\mathcal{S} \cup \{x\}) \geq \mu(\mathcal{S})$.*

**Corollary C.3** (Dominance). *If a chemical space measure $\mu$ is subadditive, then for any two molecular sets $\forall \mathcal{S}_1, \mathcal{S}_2 \subseteq \mathcal{U}$, if $\mathcal{S}_1 \subseteq \mathcal{S}_2$, then $\mu(\mathcal{S}_1) \leq \mu(\mathcal{S}_2)$.*

A subadditive chemical space measure is a special case of *outer measures* in the context of mathematical measure theory. *Outer measures* are relaxations of *measures*, where the latter requires a stricter additivity property, *i.e.*, $\mu(\mathcal{S}_1 \cup \mathcal{S}_2) = \mu(\mathcal{S}_1) + \mu(\mathcal{S}_2)$ for any two disjoint sets $\mathcal{S}_1, \mathcal{S}_2$ (Halmos, 2013). We consider additivity to be too strong and can conflict with intuitions in drug discovery: an additive measure defined on a discrete space must take the form of $\mu(\mathcal{S}) = \sum_{x \in \mathcal{S}} w(x)$, in which $w(\cdot)$ is a weight function that independently assigns a score to each element in the space, meaning additive measures defined on the discrete chemical space cannot capture the interrelationship between molecules in a candidate set, which counters the reality, thus can hardly tell the variety of compounds and how much of the chemical space is been covered.

# D  PROOFS FOR MONOTONICITY, SUBADDITIVITY AND DISSIMILARITY

In Figure 1 we show the chemical space measures according to whether they will satisfy monotonicity, subadditivity, and/or dissimilarity. In this section, we provide proofs to verify each measure's monotonicity, subadditivity, and dissimilarity.

Note that for aggregation-based measures, when $|\mathcal{S}| = 1$, $\mu(\mathcal{S})$ is not defined. So without loss of generality, we assume $\mu(\{x\}) = w(x)$, $x \in \mathcal{U}$ in such cases, where $w : \mathcal{U} \to \mathbb{R}$ is an importance function.

**Proposition D.1** (Monotonicity of chemical space measures.). *Reference-based chemical space measures, SumDiversity, Diameter, SumDiameter, and #Circles are monotonic. Diversity, Bottleneck, SumBottleneck, and DPP are not monotonic.*

*Proof.* In the proof, for two arbitrary molecular sets $\mathcal{S}_1$ and $\mathcal{S}_2$ as stated in the monotonicity axiom, we consider combining them into $\mathcal{S} := \mathcal{S}_1 \cup \mathcal{S}_2$.

For the **reference-based** chemical space measures, We prove the monotonicity and subadditivity simultaneously for reference-based measures by first proving the monotonicity and subadditivity of the maximum of $\text{cover}(\cdot, \cdot)$.

To prove the monotonicity and subadditivity of the maximum of $\text{cover}(\cdot, \cdot)$, we consider combining any two molecular sets $\mathcal{S}_1, \mathcal{S}_2 \subseteq \mathcal{U}$. For any reference $y \in \mathcal{R}$, we have

$$\max \left( \max_{x \in \mathcal{S}_1} \text{cover}(x, y), \max_{x \in \mathcal{S}_2} \text{cover}(x, y) \right)$$
$$\leq \max_{x \in \mathcal{S}_1 \cup \mathcal{S}_2} \text{cover}(x, y)$$
$$\leq \max_{x \in \mathcal{S}_1} \text{cover}(x, y) + \max_{x \in \mathcal{S}_2} \text{cover}(x, y).$$

Therefore,

$$\max \left( \text{Coverage}(\mathcal{S}_1), \text{Coverage}(\mathcal{S}_2) \right)$$
$$\leq \text{Coverage}(\mathcal{S}_1 \cup \mathcal{S}_2)$$
$$\leq \text{Coverage}(\mathcal{S}_1) + \text{Coverage}(\mathcal{S}_1),$$

thus proving the Coverage measure is monotonic and subadditive.

For the **SumDiversity** chemical space measure, note that the monotonicity corollary (Corollary C.2) can also entail the monotonicity property defined in Axiom 4.1 (by sequentially adding molecules from $\mathcal{S}_2$ to $\mathcal{S}_1$ or from $\mathcal{S}_1$ to $\mathcal{S}_2$). So we prove the monotonicity (defined in Axiom 4.1) by proving

the monotonicity [single molecule] (defined in C.2). For a molecule $x \in \mathcal{U}$ and a molecular set $\mathcal{S} \subseteq \mathcal{U}$ with size $n$, using the fact that $\text{SumDiversity}(\mathcal{S}) = n \cdot \text{Diversity}(\mathcal{S})$, we have

$$\text{SumDiversity}(\mathcal{S} \cup \{x\}) - \text{SumDiversity}(\mathcal{S})$$

$$= (n-1) \cdot \text{Diversity}(\mathcal{S}) + \frac{2}{n} \sum_{y \in \mathcal{S}} d(x, y) - n \cdot \text{Diversity}(\mathcal{S})$$

$$= -\text{Diversity}(\mathcal{S}) + \frac{2}{n} \sum_{y \in \mathcal{S}} d(x, y).$$

Comparing $\text{Diversity}(\mathcal{S})$ and $\frac{2}{n} \sum_{y \in \mathcal{S}} d(x, y)$ is equivalent to comparing $\sum_{y_1 \neq y_2 \in \mathcal{S}} d(y_1, y_2)$ and $(n-1) \sum_{y \in \mathcal{S}} d(x, y)$. Consider each "triangle" tuple $(y_1, y_2, x)$, due to the metric characteristics of $d$, we have $d(y_1, y_2) \leq d(x, y_1) + d(x, y_2)$, meaning that $\sum_{y_1 \neq y_2 \in \mathcal{S}} d(y_1, y_2)$ is less or equal to $(n-1) \sum_{y \in \mathcal{S}} d(x, y)$. When $w(x) = 0$, $\text{SumDiversity}(\mathcal{S} \cup \{x\}) - \text{SumDiversity}(\mathcal{S}) \geq 0$, proving the monotonicity of SumDiversity.

For the **Diameter** chemical space measure, when the importance function satisfies $w(x) = 0, \forall x \in \mathcal{U}$, we have

$$\text{Diameter}(\mathcal{S}) = \max_{\substack{x, y \in \mathcal{S} \\ x \neq y}} d(x, y)$$

$$\geq \max \left( \max_{\substack{x, y \in \mathcal{S}_1 \\ x \neq y}} d(x, y), \max_{\substack{x, y \in \mathcal{S}_2 \\ x \neq y}} d(x, y) \right)$$

$$= \max(\text{Diameter}(\mathcal{S}_1), \text{Diameter}(\mathcal{S}_2)),$$

proving the monotonicity of Diameter.

For the **SumDiameter** chemical space measure, when the importance function satisfies $w(x) = 0, \forall x \in \mathcal{U}$, we have

$$\text{SumDiameter}(\mathcal{S}) = \sum_{x \in \mathcal{S}} \max_{\substack{y \in \mathcal{S} \\ y \neq x}} d(x, y)$$

$$\geq \max \left( \sum_{x \in \mathcal{S}_1} \max_{\substack{y \in \mathcal{S}_1 \\ y \neq x}} d(x, y), \sum_{x \in \mathcal{S}_2} \max_{\substack{y \in \mathcal{S}_2 \\ y \neq x}} d(x, y) \right)$$

$$= \max(\text{SumDiameter}(\mathcal{S}_1), \text{SumDiameter}(\mathcal{S}_2)),$$

proving the monotonicity of SumDiameter.

For the **#Circles** chemical space measure, we define $\mathcal{C}^*(\mathcal{S}) \subseteq \mathcal{S}$ as an arbitrary set that satisfies $|\mathcal{C}^*(\mathcal{S})| = \text{\#Circles}(\mathcal{S})$. For any two molecular sets $\mathcal{S}_1, \mathcal{S}_2 \subseteq \mathcal{U}$, since $\mathcal{C}^*(\mathcal{S}_1) \subseteq \mathcal{S}_1 \cup \mathcal{S}_2$, according to the definition of #Circles, we have

$$\text{\#Circles}(\mathcal{S}_1) \leq \text{\#Circles}(\mathcal{S}_1 \cup \mathcal{S}_2).$$

Similarly, we also have

$$\text{\#Circles}(\mathcal{S}_2) \leq \text{\#Circles}(\mathcal{S}_1 \cup \mathcal{S}_2),$$

proving the monotonicity of #Circles.

For the **Diversity** chemical space measure, we disprove its monotonicity by proving it violates the monotonicity [single molecule] corollary (Corollary C.2). For a molecule $x \in \mathcal{U}$ and a molecular set $\mathcal{S} \subseteq \mathcal{U}$ with size $n > 1$, if $x \notin \mathcal{S}$, we have

$$\text{Diversity}(\mathcal{S} \cup \{x\})$$

$$= \frac{2}{(n+1)n} \left[ \sum_{\substack{y,y' \in \mathcal{S} \\ y \neq y'}} d(y,y') + \sum_{y \in \mathcal{S}} d(x,y) \right]$$

$$= \frac{n-1}{n+1} \cdot \text{Diversity}(\mathcal{S}) + \frac{2}{(n+1)n} \sum_{y \in \mathcal{S}} d(x,y).$$

And the change in Diversity is

$$\text{Diversity}(\mathcal{S} \cup \{x\}) - \text{Diversity}(\mathcal{S})$$

$$= \left( \frac{n-1}{n+1} - \frac{n+1}{n+1} \right) \text{Diversity}(\mathcal{S}) + \frac{2}{(n+1)n} \sum_{y \in \mathcal{S}} d(x,y)$$

$$= \frac{2}{n+1} \left[ -\text{Diversity}(\mathcal{S}) + \frac{1}{n} \sum_{y \in \mathcal{S}} d(x,y) \right].$$

When the average distance of $x$ and $\mathcal{S}$, *i.e.*, $\frac{1}{n} \sum_y d(x,y)$, is less than Diversity$(\mathcal{S})$ (*e.g.*, adding a molecule on the "segment" between two existing molecules), Diversity would decrease, thus violating the monotonicity corollary and proving Diversity is not monotonic.

For the **Bottleneck** chemical space measure, we disprove its subadditivity by proving it violates the monotonicity [single molecule] corollary (Corrolary C.2). Consider adding a molecule $x$ into a molecular set $\mathcal{S}$ with size $n > 1$. If $x \notin \mathcal{S}$, we have

$$\text{Bottleneck}(\mathcal{S} \cup \{x\}) = \min\left( \text{Bottleneck}(\mathcal{S}), \ \min_{y \in \mathcal{S}} d(x,y) \right).$$

When $x$ introduces a more restricting bottleneck, *i.e.*, $\min_y d(x,y) < \text{Bottleneck}(\mathcal{S})$, we will have Bottleneck$(\mathcal{S} \cup \{x\}) < $ Bottleneck$(\mathcal{S})$, violating the monotonicity corollary, thus proving Bottleneck is not monotonic.

For the **SumBottleneck** chemical space measure, we disprove its monotonicity by proving it violates the monotonicity [single molecule] corollary (Corrolory C.2). Consider adding a molecule $x$ into a molecular set $\mathcal{S}$ with size $n > 1$. If $x \notin \mathcal{S}$, we have

$$\text{SumBottleneck}(\mathcal{S} \cup \{x\})$$

$$= \sum_{y \in \mathcal{S}} \min\left( \min_{\substack{y' \in \mathcal{S} \\ y' \neq y}} d(y,y'), \ d(x,y) \right) + \min_{y \in \mathcal{S}} d(x,y),$$

and

$$\text{SumBottleneck}(\mathcal{S}) = \sum_{y \in \mathcal{S}} \min_{\substack{y' \in \mathcal{S} \\ y' \neq y}} d(y,y').$$

When $x$ introduces some more restricting bottlenecks, *i.e.*, for many $y \in \mathcal{S}$, $d(x,y)$ is small (*e.g.*, adding a molecule into a set whose size is two, and the new molecule is added near one of the two

molecules), we will have SumBottleneck$(\mathcal{S} \cup \{x\}) <$ SumBottleneck$(\mathcal{S})$, violating the monotonicity corollary, thus proving SumBottleneck is not subadditive.

For the **DPP** chemical space measure, we disprove its monotonicity by proving it violates the monotonicity [single molecule] corollary (Corrolory C.2). Consider adding $x$ into $\{x_0\}$ where $x \neq x_0$ $1 - d(x, x_0)$ is denoted as $b$. We have

$$\text{DPP}(\{x_0, x\}) = \begin{vmatrix} 1 & b \\ b & 1 \end{vmatrix} = 1 - b^2.$$

When $b > 0$ we will have DPP$(\{x_0, x\}) <$ DPP$(\{x_0\}) = 1$, violating the monotonicity corollary, thus proving DPP is not monotonic.

$\square$

**Proposition D.2** (Subadditivity of chemical space measures.). *Reference-based chemical space measures, Bottleneck, SumBottleneck, and #Circles are subadditive. Diversity, SumDiversity, Diameter, and SumDiameter are not subadditive.*

*Proof.* For the **reference-based** chemical space measures, the subadditivity is provided under Proposition D.1.

For the **Bottleneck** chemical space measure, when the importance function satisfies $w(x) \geq M$, $\forall x \in \mathcal{U}$, where $M$ is the upper bound of Bottlenkeck (*e.g.*, $w(x) = \infty$), we have

$$\text{Bottleneck}(\mathcal{S}) = \min_{\substack{x,y \in \mathcal{S} \\ x \neq y}} d(x,y)$$

$$\leq \min \left( \min_{\substack{x,y \in \mathcal{S}_1 \\ x \neq y}} d(x,y), \min_{\substack{x,y \in \mathcal{S}_2 \\ x \neq y}} d(x,y) \right)$$

$$\leq \text{Bottleneck}(\mathcal{S}_1) + \text{Bottleneck}(\mathcal{S}_2)),$$

proving the subadditivity of Bottleneck.

For the **SumBottleneck** chemical space measure, when the importance function satisfies $w(x) \geq M$, $\forall x \in \mathcal{U}$, where $M$ is the upper bound of SumBottleneck (*e.g.*, $w(x) = \infty$), we have

$$\text{SumBottleneck}(\mathcal{S}) = \sum_{x \in \mathcal{S}_1 \cup \mathcal{S}_2} \min_{\substack{y \in \mathcal{S}_1 \cup \mathcal{S}_2 \\ y \neq x}} d(x,y)$$

$$= \sum_{x \in \mathcal{S}_1} \min_{\substack{y \in \mathcal{S}_1 \cup \mathcal{S}_2 \\ y \neq x}} d(x,y) + \sum_{x \in \mathcal{S}_2} \min_{\substack{y \in \mathcal{S}_1 \cup \mathcal{S}_2 \\ y \neq x}} d(x,y)$$

$$\leq = \sum_{x \in \mathcal{S}_1} \min_{\substack{y \in \mathcal{S}_1 \\ y \neq x}} d(x,y) + \sum_{x \in \mathcal{S}_2} \min_{\substack{y \in \mathcal{S}_2 \\ y \neq x}} d(x,y)$$

$$= \text{SumBottleneck}(\mathcal{S}_1) + \text{SumBottleneck}(\mathcal{S}_2)),$$

proving the subadditivity of SumBottleneck.

For the **#Circles** chemical space measure, we prove its subadditivity by contradiction. For any two molecular sets $\mathcal{S}_1, \mathcal{S}_2 \subseteq \mathcal{U}$, we assume #Circles$(\mathcal{S}_1 \cup \mathcal{S}_2) >$ #Circles$(\mathcal{S}_1) +$ #Circles$(\mathcal{S}_1)$. Use the notations $\mathcal{C}_1 := \mathcal{C}^*(\mathcal{S}_1 \cup \mathcal{S}_2) \cap \mathcal{S}_1$ and $\mathcal{C}_2 := \mathcal{C}^*(\mathcal{S}_1 \cup \mathcal{S}_2) \cap \mathcal{S}_2$. We have

$$|\mathcal{C}_1| + |\mathcal{C}_2| \geq \text{\#Circles}(\mathcal{S}_1 \cup \mathcal{S}_2) > \text{\#Circles}(\mathcal{S}_1) + \text{\#Circles}(\mathcal{S}_1).$$

Since all values are non-negative, we must have $|\mathcal{C}_1| >$ #Circles$(\mathcal{S}_1)$ or $|\mathcal{C}_2| >$ #Circles$(\mathcal{S}_2)$, contradicting with the definition of #Circles$(\mathcal{S}_1)$ or #Circles$(\mathcal{S}_1)$, thus proving the subadditivity of #Circles.

For the **Diversity** and **SumDiversity** chemical space measures, we disprove their subadditivity by providing a counter-example. For two disjoint molecular sets with two molecules in each, *i.e.*, $\{x_1, x_2\}$ and $\{x_3, x_4\}$, we denote $d_{ij} = d(x_i, x_j)$. Then we have

$$
\begin{aligned}
&\text{Diversity}(\{x_1, x_2, x_3, x_4\}) \\
&= \text{Diversity}(\{x_1, x_2, x_3, x_4\}) \\
&= \frac{2}{4 \cdot 3}(d_{12} + d_{13} + d_{14} + d_{23} + d_{24} + d_{34}),
\end{aligned}
$$

and

$$
\text{Diversity}(\{x_1, x_2\}) + \text{Diversity}(\{x_3, x_4\}) = d_{12} + d_{34}.
$$

Similarly,

$$
\begin{aligned}
&\text{SumDiversity}(\{x_1, x_2, x_3, x_4\}) \\
&= 4 \cdot \text{Diversity}(\{x_1, x_2, x_3, x_4\}) \\
&= 4 \cdot \frac{2}{4 \cdot 3}(d_{12} + d_{13} + d_{14} + d_{23} + d_{24} + d_{34}),
\end{aligned}
$$

and

$$
\text{SumDiversity}(\{x_1, x_2\}) + \text{SumDiversity}(\{x_3, x_4\}) = 2 \cdot d_{12} + 2 \cdot d_{34}.
$$

When the inter-set distances are larger than the inner-set distances, *i.e.*, $d_{13}+d_{14}+d_{23}+d_{24} > 5\cdot(d_{12}+d_{34})$, we will have $\text{Diversity}(\{x_1, x_2, x_3, x_4\}) > \text{SumDiversity}\{x_1, x_2\} + \text{Diversity}\{x_3, x_4\}$. Similarly, when $d_{13} + d_{14} + d_{23} + d_{24} > 2 \cdot (d_{12} + d_{34})$, we will have $\text{SumDiversity}(\{x_1, x_2, x_3, x_4\}) > \text{SumDiversity}\{x_1, x_2\} + \text{SumDiversity}\{x_3, x_4\}$, thus proving both Diversity and SumDiversity measures are not subadditive.

For the **Diameter** chemical space measure, For two disjoint molecular sets $\mathcal{S}_1, \mathcal{S}_2 \subseteq \mathcal{U}$ whose sizes are larger than one, we have

$$
\text{Diameter}(\mathcal{S}_1 \cup \mathcal{S}_2) = \max_{\substack{x,y \in \mathcal{S}_1 \cup \mathcal{S}_2 \\ x \neq y}} d(x, y),
$$

and

$$
\text{Diameter}(\mathcal{S}_1) + \text{Diameter}(\mathcal{S}_2) = \max_{\substack{x,x' \in \mathcal{S}_1 \\ x \neq x'}} d(x, x') + \max_{\substack{y,y' \in \mathcal{S}_2 \\ y \neq y'}} d(y, y').
$$

When the maximum inter-set distance is larger than the maximum inner-set distance, *i.e.*, $\max_{x,y,\in\mathcal{S}_1\cup\mathcal{S}_2} d(x, y) > \max_{x,x',\in\mathcal{S}_1} d(x, x') + \max_{y,y',\in\mathcal{S}_2} d(y, y')$, we will have $\text{Diameter}(\mathcal{S}_1 \cup \mathcal{S}_2) > \text{Diameter}(\mathcal{S}_1) + \text{Diameter}(\mathcal{S}_2)$, thus proving the Diameter measure is not subadditive.

For the **SumDiameter** chemical space measure, we disprove its subadditivity by providing a counter-example. For two disjoint molecular sets with two molecules in each, *i.e.*, $\{x_1, x_2\}$ and $\{x_3, x_4\}$, we denote $d_{ij} = d(x_i, x_j)$. Then we have

$$\text{SumDiversity}(\{x_1, x_2, x_3, x_4\}) = \sum_{\substack{i \in [4]}} \max_{\substack{j \in [4] \\ j \neq i}} d(x_i, x_j),$$

and

$$\text{SumDiversity}(\{x_1, x_2\}) + \text{SumDiversity}(\{x_3, x_4\}) = 2 \cdot d_{12} + 2 \cdot d_{34}.$$

When the inter-set distances are larger than the inner-set distances, *i.e.*, $d_{13}, d_{14}, d_{23}, d_{24} > d_{12}, d_{34}$, we will have $\text{SumDiversity}(\{x_1, x_2, x_3, x_4\}) > \text{SumDiversity}\{x_1, x_2\} + \text{SumDiversity}\{x_3, x_4\}$, thus proving the SumDiameter measure is not subadditive.

$\square$

**Proposition D.3** (Dissimilarity property of chemical space measures.)**.** *Diversity, SumDiversity, Diameter, SumDiameter, Bottleneck, SumBottleneck, DPP, and #Circles have preferences to dissimilarity.*

*Proof.* In the proof, for the three molecules $x_0, x_1, x_2 \in \mathcal{U}$ as stated in the dissimilarity axiom in Section 4.1, we assume $d(x_0, x_1) \geq d(x_0, x_2)$.

For the **Diversity** chemical space measure, we have

$$\text{Diversity}(\{x_0, x_1\}) = d(x_0, x_1) \geq d(x_0, x_2) = \text{Diversity}(\{x_0, x_2\}),$$

proving the dissimilarity property of Diversity.

For the **SumDiversity** chemical space measure, we have

$$\text{SumDiversity}(\{x_0, x_1\}) = 2 \cdot d(x_0, x_1) \geq 2 \cdot d(x_0, x_2) = \text{SumDiversity}(\{x_0, x_2\}),$$

proving the dissimilarity property of SumDiversity.

For the **Diameter** chemical space measure, we have

$$\text{Diameter}(\{x_0, x_1\}) = d(x_0, x_1) \geq d(x_0, x_2) = \text{Diameter}(\{x_0, x_2\}),$$

proving the dissimilarity property of Diameter.

For the **SumDiameter** chemical space measure, we have

$$\text{SumDiameter}(\{x_0, x_1\}) = 2 \cdot d(x_0, x_1) \geq 2 \cdot d(x_0, x_2) = \text{SumDiameter}(\{x_0, x_2\}),$$

proving the dissimilarity property of SumDiameter.

For the **Bottleneck** chemical space measure, we have

$$\text{Bottleneck}(\{x_0, x_1\}) = d(x_0, x_1) \geq d(x_0, x_2) = \text{Bottleneck}(\{x_0, x_2\}),$$

proving the dissimilarity property of Bottleneck.

For the **SumBottleneck** chemical space measure, we have

$$\text{SumBottleneck}(\{x_0, x_1\}) = 2 \cdot d(x_0, x_1) \geq 2 \cdot d(x_0, x_2) = \text{SumBottleneck}(\{x_0, x_2\}),$$

proving the dissimilarity property of SumBottleneck.

For the **DPP** chemical space measure defined with the Tanimoto similarity, denoting $d(x_0, x_1)$ and $d(x_0, x_2)$ as $d_1$ and $d_2$ respectively, when $d_1, d_2 \in [0, 1]$ (Tanimoto distances),

we have

$$\text{DPP}(\{x_0, x_1\}) = \begin{vmatrix} 1 & 1 - d_1 \\ 1 - d_1 & 1 \end{vmatrix} = 2d_1 - d_1^2 \geq 2d_2 - d_2^2 = \begin{vmatrix} 1 & 1 - d_2 \\ 1 - d_2 & 1 \end{vmatrix} = \text{DPP}(\{x_0, x_2\}),$$

proving the dissimilarity property of DPP.

For the **#Circles** chemical space measure, we have

$$\text{#Circles}(\{x_0, x_1\}) = 1 + \mathbb{I}[d(x_0, x_1) > t] \geq 1 + \mathbb{I}[d(x_0, x_2) > t] = \text{#Circles}(\{x_0, x_2\}),$$

proving the dissimilarity property of #Circles.

$\square$

**Discussion on the dissimilarity property of reference-based measures.** For a reference-based measure, considering adding a new molecule $x_1$ or $x_2$ to the existing moelculer set $\mathcal{S} = \{x_0\}$, we have

$$\text{Coverage}(\{x_0, x_1\}, \mathcal{R}) = \sum_{y \in \mathcal{R}} \left( \max(\text{cover}(x_0, y), \text{cover}(x_1, y)) \right),$$

$$\text{Coverage}(\{x_0, x_2\}, \mathcal{R}) = \sum_{y \in \mathcal{R}} \left( \max(\text{cover}(x_0, y), \text{cover}(x_2, y)) \right),$$

where the values of $\text{Coverage}(\{x_0, x_1\}, \mathcal{R})$ and $\text{Coverage}(\{x_0, x_2\}, \mathcal{R})$ will depend on the particular definition of $\text{cover}(\cdot, \cdot)$ and the choice of the reference set $\mathcal{R}$.

Generally, an arbitrary coverage function and an arbitrary reference set do not necessarily meet the dissimilarity requirement. In our study, we define the cover function as $\text{cover}(x, y) :=$ $\mathbb{I}[\text{molecule } x \text{ contains fragment } y]$, where $\mathbb{I}[\cdot]$ is the indicator function, and $y$ is a fragment contained in the reference set $\mathcal{R}$. Some counter-examples can be easily constructed, for instance, if $x_1$ is far away from the reference set $\mathcal{R}$ while $x_2$ is very close to $\mathcal{R}$ (even contained in $\mathcal{R}$), then the Coverage measure will prefer the molecule $x_2$ instead of the more dissimilar molecule $x_1$.

Therefore, the dissimilarity property does not hold for chemical space measures #FG, #RS, and #BM.

**Discussion on a new definition of reference-based measure.** In addition to counting the number of fragments in the reference set $\mathcal{R}$ contained in the molecular set $\mathcal{S}$, another way to define a reference-based measure is to investigate the distance from $\mathcal{S}$ to $\mathcal{R}$. Particularly, we could define such a reference-based measure as below:

$$\text{Coverage}(\mathcal{S}, \mathcal{R}; d, t) := \sum_{y \in \mathcal{R}} \mathbb{I}[\exists x \in \mathcal{S} \text{ such that } d(x, y) < t], \tag{10}$$

where $\mathbb{I}[\cdot]$ is the indicator function, $d$ is the distance metric, and $t$ is the distance threshold.

Similar to the discussion on #FG, #RS, and #BM, the characteristics of the reference-based measure defined by Eq. 10 highly relies on the property of $\mathcal{R}$, and an arbitrary reference set $\mathcal{R}$ can not guarantee the dissimilarity property of this measure.

We now prove that the reference-based measure defined by Eq. 10 satisfies both monotonicity and subadditivity. In the proof, $\text{Coverage}(\cdot)$ refers to the definition in Eq. 10.

*Proof.* We first prove the monotonicity and subadditivity of the indicator function $\mathbb{I}[\cdot]$, where we consider combining any two molecular sets $\mathcal{S}_1, \mathcal{S}_2 \subseteq \mathcal{U}$. For any reference $y \in \mathcal{R}$ and a distance threshold $t$, we have

$$\max\left(\mathbb{I}[\exists x \in \mathcal{S}_1 \text{ such that } d(x,y) < t],\ \mathbb{I}[\exists x \in \mathcal{S}_2 \text{ such that } d(x,y) < t]\right)$$
$$\leq \mathbb{I}[\exists x \in \mathcal{S}_1 \cup \mathcal{S}_2 \text{ such that } d(x,y) < t]$$
$$\leq \mathbb{I}[\exists x \in \mathcal{S}_1 \text{ such that } d(x,y) < t] + \mathbb{I}[\exists x \in \mathcal{S}_2 \text{ such that } d(x,y) < t].$$

Therefore,

$$\max\left(\text{Coverage}(\mathcal{S}_1), \text{Coverage}(\mathcal{S}_2)\right)$$
$$\leq \text{Coverage}(\mathcal{S}_1 \cup \mathcal{S}_2)$$
$$\leq \text{Coverage}(\mathcal{S}_1) + \text{Coverage}(\mathcal{S}_1),$$

thus proving the Coverage measure defined by Eq. 10 is monotonic and subadditive. $\square$

## E  RANDOM SUBSET EXPERIMENT DETAILS

### E.1  BIO-ACTIVITY DATASET

The 10K BioActivity dataset (Koutsoukas et al., 2014) contains 10,000 compound samples excerpted from the ChEMBL database (Gaulton et al., 2017) with bio-activity labels. These labels are the 50 largest ChEMBL activity classes, including enzymes (*e.g.*, proteases, lyases, reductases, hydrolases, and kinases) and membrane receptors (*e.g.*, GPCRs and non-GPCRs). The label distribution is shown in Figure 5.

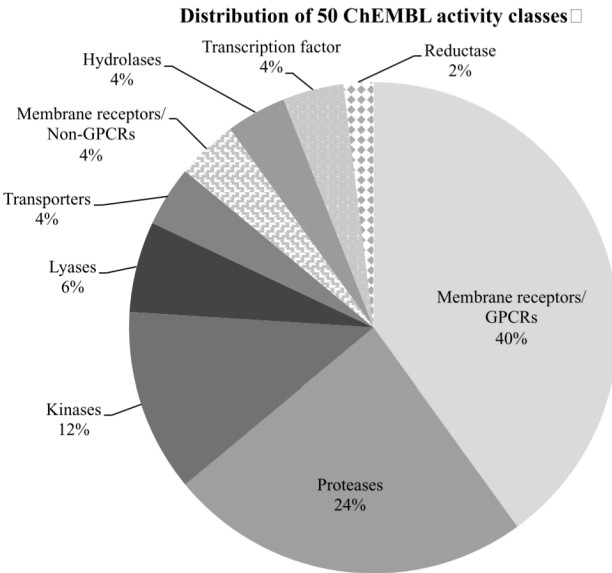

Figure 5: Label distribution of the BioActivity dataset (Koutsoukas et al., 2014). 50 bio-activity functionality classes are included.

We use UMAP (McInnes et al., 2018) to visualize the molecules in this dataset based on their Morgan fingerprints as displayed Figure 6. From the visualization, we can see that fingerprint similarity is indeed correlated with bio-activity similarity.

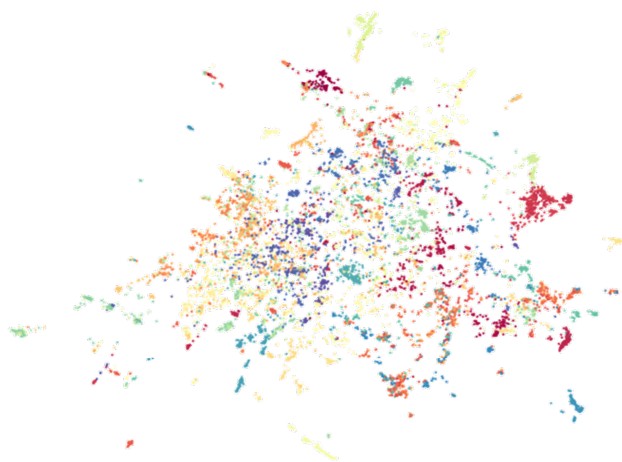

Figure 6: UMAP visualization of compounds in the BioActivity dataset. Different colors stand for different bio-activity labels.

## E.2 RANDOM SUBSETS WITH FIXED SIZES

We first consider randomly sampled molecular subsets of the BioActivity dataset with a fixed size $n$. We randomly sample $n$ molecules $\mathcal{S}$ from the dataset and compute the biological functionality coverage $GS(\mathcal{S})$ as well as each chemical space measure $\mu(\mathcal{S})$. By repeating the randomization, we can calculate Spearman's correlation between the gold standard GS and each individual measure $\mu$. We run the experiment for three different fixed sizes $n = 50$, $n = 200$, and $n = 1000$ to represent different molecular distribution density. The pair-wise correlations between measures are displayed in Figure 8.

Furthermore, when $n = 50$ and the molecules are distributed sparsely, all chemical space measures are positively correlated with the gold standard. However, when the subset size increases to $n = 200$ and $n = 1000$, Bottleneck and DPP becomes negatively correlated, while other measure tend to perform better. This is because these two measures will be bounded by the most similar molecular pair, thus severely conflicting with the subadditivity axiom.

In this experiment, we repeat Algorithm 1 for ten times to obtain reliable correlations.

---

**Algorithm 1** Calculating chemical space measures for random subsets with fixed sizes.

---

**Input:** The fixed subset size $n$; The bio-activity dataset $\{(x_i, y_i)\}_{i=1}^{10K}$ where $y_i \in \mathcal{Y}$ are bio-activity labels and $|\mathcal{Y}| = 50$; $K$ chemical space measures $\{\mu_k\}_{k=1}^{K}$.
**repeat**
    Sample a number $m$ uniformly from $\{1, \ldots, 50\}$.
    Sample $m$ labels $\mathcal{Y}'$ uniformly from $\mathcal{Y}$.
    Sample $n$ molecules $\mathcal{S}$ with labels in $\mathcal{Y}'$ uniformly.
    Compute $GS(\mathcal{S})$ and $\mu_k(\mathcal{S})$ for $k \in [K]$.
**until** repeated for 1000 times
Calculate the correlations between GS and $\{\mu_k\}_{k=1}^{K}$ based on the 1000-times experiment results.

---

**Experiment results.**    We show the experiment results for different fixed random set size $n$ in Figure 7. We find the #Circles and SumBottleneck measures perform constantly better than all other measures.

When the fixed size $n$ increases, most chemical space measures' performances also increase, except for Bottleneck and DPP, meaning they are not suitable for measuring the variety when the molecules are distributed crowdedly.

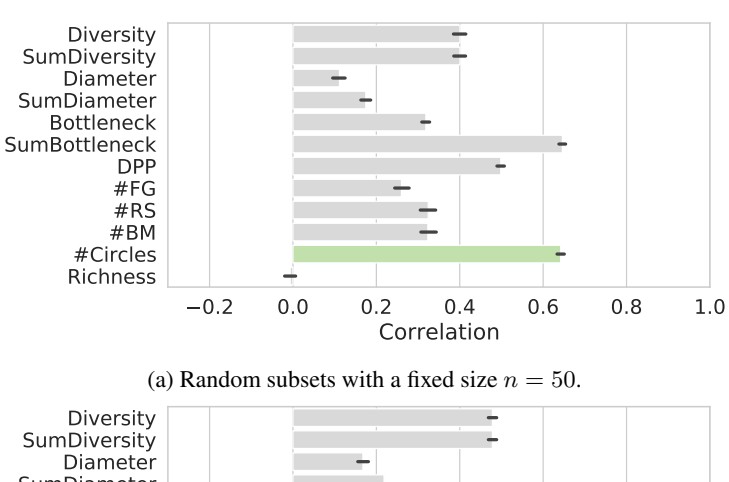

(a) Random subsets with a fixed size $n = 50$.

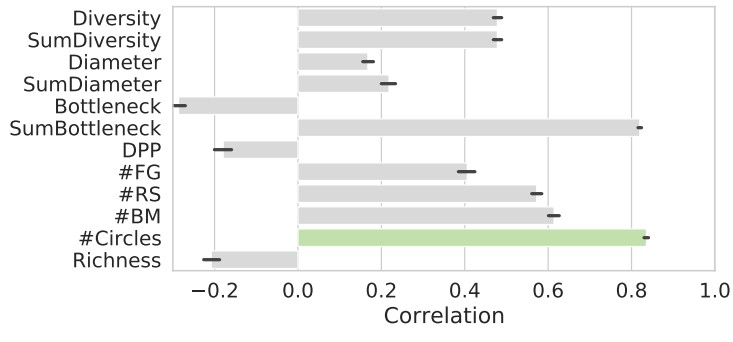

(b) Random subsets with a fixed size $n = 200$.

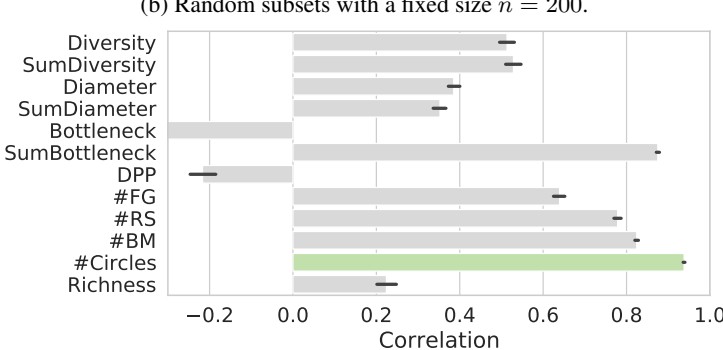

(c) Random subsets with a fixed size $n = 1000$.

Figure 7: Correlations between the gold standard GS and chemical space measures in the fixed-size random subset setting. The fixed size is set as different values. A larger correlation indicates the better. The average results are obtained by running experiments independently for ten times.

**Correlation between chemical space measures.** We also visualize the pairwise correlation between chemical space measures in Figure 8. From the figure we can see that, the gold standard GS, #Circles, and SumBottleneck are most similar with each other in the fixed-size setting.

**Threshold $t$ for #Circles.** The #Circles threshold $t$ is selected to maximize the correlation to the gold standard GS. Taking $n = 200$ as an example, we test different $t$ values as Figure 9 displays and select $t = 0.70$ as the threshold. We can see #Circles works well for a wide range of thresholds like $[0.40, 0.70]$. In Olivecrona et al. (2017), the authors suggest to use a threshold $t = 0.60$ to decide whether two molecules are dissimilar with each other[3], which aligns our results. For $n = 50$ and $n = 1000$, the threshold is set as $t = 0.70$ and $t = 0.65$ respectively.

---

[3] In the original text, the authors suggest a similarity threshold of $0.40$ that is equivalent to a distance threshold of $0.60$.

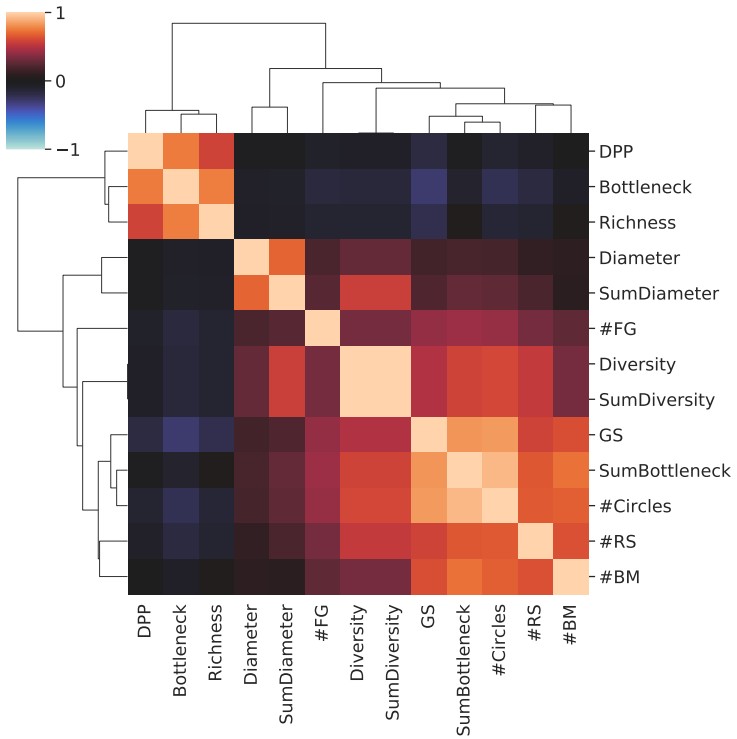

Figure 8: Correlations between chemical space measures in the fixed-size random subset setting. The fixed size is set as $n = 200$. A larger correlation indicates the better. The average results are obtained by running experiments independently for ten times.

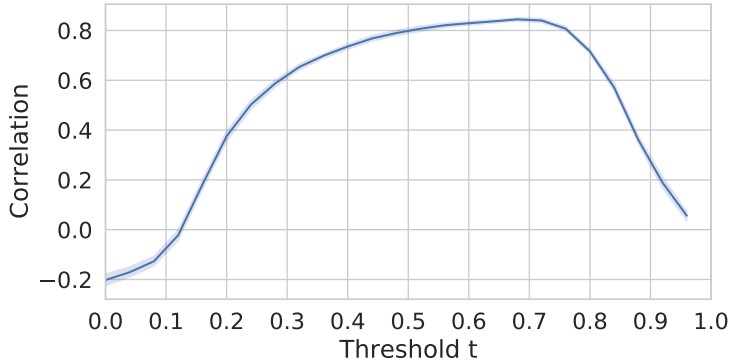

Figure 9: Correlations between the gold standard GS and the #Circles measure in the fixed-size setting with different threshold $t$. The fixed size is set as $n = 200$. A larger correlation indicates the better. The average results are obtained by running experiments independently for ten times.

**Distance metric $d$.** We also study the impact of distance metric $d$. In Table 3, we listed the experiment results for both fingerprint-based Tanimoto distance and VAE-based latent space dissimilarity (Samanta et al., 2020). We find the experiment results obtained with the VAE dissimilarity remain consistent with the results obtained with the Tanimoto distance.

### E.3 RANDOM SUBSETS WITH GROWING SIZES

To mimic the molecular generation process, we also grow the size of the subsets. Specifically, for a maximum size $n$, we sequentially sample $n$ molecules without replacement to form $n$ subsets

Table 3: Correlations between the gold standard and chemical space measures in the fixed-size random subset setting. The fixed size is set as $n = 200$. A larger correlation indicates the better. Results are obtained by averaging ten independent experiments. *Italic* texts indicate molecular representation and the distance metric. The top three measures are highlighted in green, and the best measure is printed in **bold**.

| | Aggregation-based | |
| | *Tanimoto distance* | *VAE dissimilarity* |
|---|---|---|
| Diversity | $0.478 \pm 0.011$ | $0.388 \pm 0.040$ |
| SumDiversity | $0.478 \pm 0.011$ | $0.388 \pm 0.040$ |
| Diameter | $0.179 \pm 0.031$ | $0.112 \pm 0.022$ |
| SumDiameter | $0.228 \pm 0.028$ | $0.201 \pm 0.029$ |
| Bottleneck | $-0.293 \pm 0.015$ | $-0.298 \pm 0.027$ |
| SumBottleneck | $0.821 \pm 0.010$ | $0.527 \pm 0.013$ |
| DPP | $-0.183 \pm 0.021$ | $-0.244 \pm 0.030$ |
| | **Reference-based** | |
| | *Fgragment* | |
| #FG | $0.421 \pm 0.033$ | |
| #RS | $0.574 \pm 0.025$ | |
| #BM | $0.610 \pm 0.028$ | |
| | **Locality-based** | |
| | *Tanimoto distance* | *VAE dissimilarity* |
| **#Circles** | $\mathbf{0.831 \pm 0.008}$ | $0.745 \pm 0.014$ |
| | *SMILES* | |
| Richness | $-0.207 \pm 0.025$ | |

$\{\mathcal{S}_i = \{x_1, \dots, x_i\}\}_{i=1}^n$. For both the gold standard GS and a chemical space measure $\mu$, we record their values as $\mathcal{S}$ grows into a time series. , *e.g.*, $\{(i, \mu(\mathcal{S}_i))\}_{i=1}^n$. Comparing the trajectory of a chemical space measure with the trajectory of the gold standard, we can observe which measure behaves more similarly to GS. We quantitatively estimate the similarity of their trajectories with the dynamic time warping (DTW) distance of the two time series.

In this experiment, we repeat Algorithm 2 for ten times to obtain reliable DTW distances.

---

**Algorithm 2** Calculating chemical space measures for random subsets with growing sizes.

---

**Input:** The maximum subset size $n$; The bio-activity dataset $\{(x_i, y_i)\}_{i=1}^{10K}$ where $y_i \in \mathcal{Y}$ are bio-activity labels and $|\mathcal{Y}| = 50$; $K$ chemical space measures $\{\mu_k\}_{k=1}^K$.
Sample a number $m$ uniformly from $\{1, \dots, 50\}$.
Sample $m$ labels $\mathcal{Y}'$ from $\mathcal{Y}$.
**for** $i$ **in** $\{1, \dots, n\}$ **do**
    Sample an unseen molecule $x_i$ whose label is in $\mathcal{Y}'$.
    Set $\mathcal{S}_i := \{x_1, \dots, x_i\}$
    Compute $\text{GS}(\mathcal{S}_i)$ and $\mu_k(\mathcal{S}_i)$ for $k \in [K]$.
**end for**
Plot chemical space measure curves for GS and $\{\mu_k\}_{k=1}^K$ where the x axes are $i \in \{1, \dots, n\}$ and the y axes are chemical space measure values $\text{GS}(\mathcal{S}_i)$ and $\mu_k(\mathcal{S}_i)$.
Transform the cumulative curves into incremental ones.
Calculate DTW distances between incremental curves.

---

**Experiment results.** To mimic the way in which generation models propose new molecules, in Algorithm 2, we require the newly sampled molecule $x_i$ to be similar to the already sampled molecules $\{x_1, \dots, x_{i-1}\}$. The specific implementation can be found in our code[4]. Moreover, we also test the following two cases: (1) All molecules are sampled uniformly; (2) The newly sampled molecule $x_i$ have to be most similar to the already sampled molecules $\{x_1, \dots, x_{i-1}\}$. The results of DTW distances for these two cases are shown in Figure 10.

---

[4]The code will be released after publication.

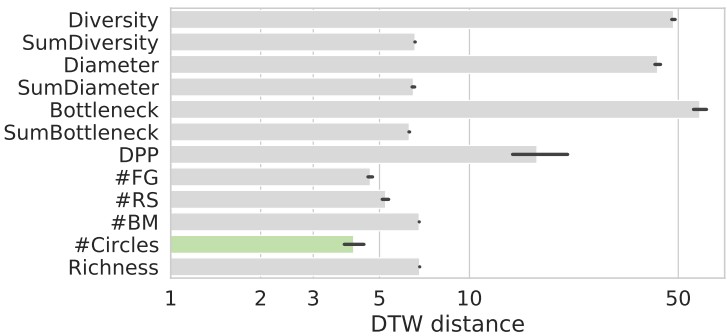

(a) The new molecule $x_i$ is uniformly sampled from all unseen molecules.

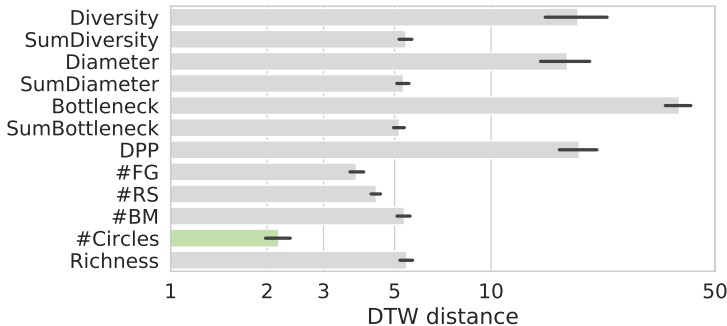

(b) The new molecule $x_i$ needs to be similar to the already sampled ones $x_1, \ldots, x_{i=1}$.

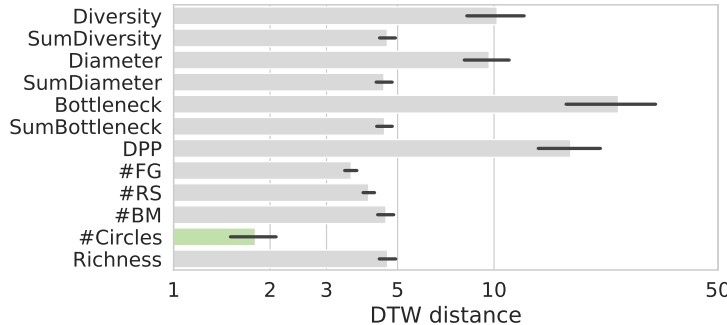

(c) The new molecule $x_i$ needs to be most similar to the already sampled ones $x_1, \ldots, x_{i=1}$.

Figure 10: DTW distances between the gold standard GS and chemical space measures in the growing-size random subset setting. The maximum size is set as $n = 1000$. A smaller distance indicates the better. The average results are obtained by running experiments independently for ten times.

From Figure 10 we can see that, #Circles performs the best. Also, as the new molecules to add become more similar to the existing ones, the advantage of #Circles over other measures becomes larger. This makes #Circles especially suitable for measuring molecular generation models.

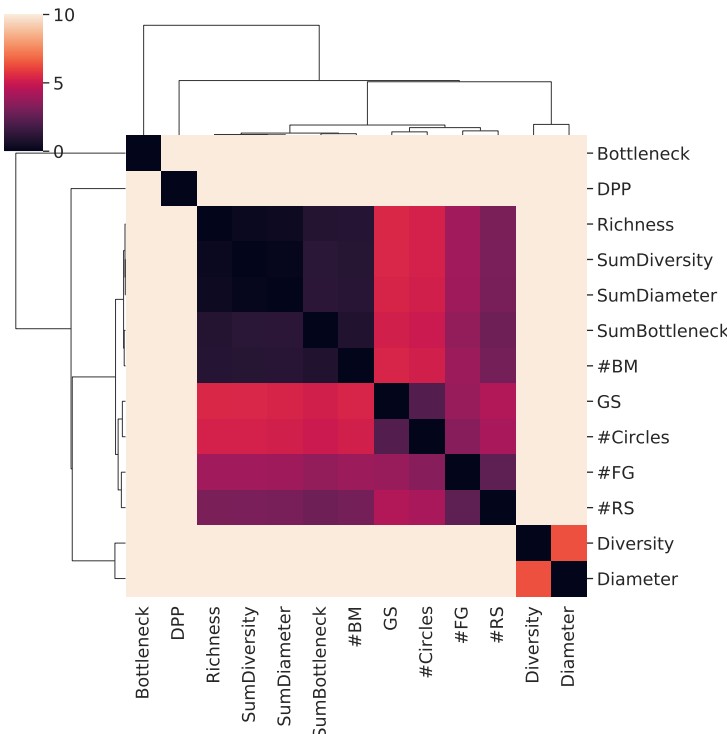

Figure 11: DTW distances between chemical space measures in the growing-size random subset setting. The maximum size is set as $n = 1000$, and the new molecule needs to be similar to the already samples ones. A smaller distance indicates the better. The average results are obtained by running experiments independently for ten times.

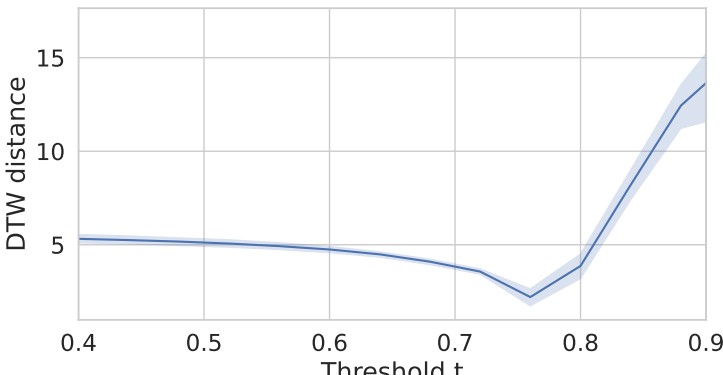

Figure 12: DTW distances between the gold standard GS and the #Circles measure in the growing-size random subset setting with different threshold $t$. The maximum size is set as $n = 1000$, and the new molecule needs to be similar to the already samples ones. A smaller distance indicates the better. The average results are obtained by running experiments independently for ten times.

**DTW distances between chemical space measures.** We visualize the pairwise DTW distances between chemical space measures in Figure 11. From the figure we can see that, the gold standard GS and the #Circles measure are most similar with each other in the growing-size setting.

In addition, we find the Richness, SumDiversity, SumDiameter, SumBottleneck, and #BM are forming a large cluster, while #FG and #RS tend to be similar with each other.

**Threshold $t$ for #Circles.** The #Circles threshold $t$ is selected to minimize the DTW distance to the gold standard GS. Taking the second scenario as an example, we test different $t$ values as Figure 12 displays and select $t = 0.76$ as the threshold. For the first and the third scenarios, the threshold is set as $t = 0.84$ and $t = 0.78$ respectively.

**Distance metric $d$.** We also study the impact of distance metric $d$. In Table 4, we listed the experiment results for both fingerprint-based Tanimoto distance and VAE-based latent space dissimilarity (Samanta et al., 2020). We find the experiment results obtained with the VAE dissimilarity remain consistent with the results obtained with the Tanimoto distance.

Table 4: DTW distances between the gold standard and chemical space measures in the growing-size random subset setting. The maximum size is set as $n = 1000$, and the new molecule needs to be similar to the already samples ones. A smaller distance indicates the better.

|  | **Aggregation-based** | |
| --- | --- | --- |
|  | *Tanimoto distance* | *VAE dissimilarity* |
| Diversity | $18.668 \pm 6.973$ | $30.063 \pm 4.284$ |
| SumDiversity | $5.425 \pm 0.404$ | $5.484 \pm 0.296$ |
| Diameter | $17.299 \pm 4.801$ | $28.071 \pm 3.917$ |
| SumDiameter | $5.328 \pm 0.396$ | $5.472 \pm 0.297$ |
| Bottleneck | $38.668 \pm 5.769$ | $37.168 \pm 5.422$ |
| SumBottleneck | $5.167 \pm 0.353$ | $5.432 \pm 0.293$ |
| DPP | $18.845 \pm 3.962$ | $12.052 \pm 2.176$ |
|  | **Reference-based** | |
|  | *Fgragment* | |
| #FG | $3.797 \pm 0.295$ | |
| #RS | $4.382 \pm 0.247$ | |
| #BM | $5.365 \pm 0.396$ | |
|  | **Locality-based** | |
|  | *Tanimoto distance* | *VAE dissimilarity* |
| **#Circles** | $\mathbf{2.173 \pm 0.910}$ | $2.470 \pm 0.629$ |
|  | *SMILES* | |
| Richness | $5.454 \pm 0.347$ | |

# F MEASURING MOLECULAR DATABASES

## F.1 MOLECULAR DATABASES

We measure the chemical space coverage for five molecular databases that are commonly used in virtual screening and generative model training:

(1) **ZINC-250k**[5] is a random subset of the ZINC database (Irwin & Shoichet, 2005) and consists of 249K commercially-available compounds from different vendors for virtual screening;

(2) **MOSES** (Polykovskiy et al., 2020) is another sub-collection of ZINC molecules. It contains approximately 2M molecules in total, filtered by molecular weights (ranged range from 250 to 350 Daltons), the number of rotatable bonds (not greater than 7), water-octanol partition coefficient (logP, less or equal than 3.5), atom types (C, N, S, O, F, Cl, Br, and H), ring cycle sizes (no larger than 8), medicinal chemistry filters (MCFs), and PAINS filters.

(3) **ChEMBL** (Gaulton et al., 2017) is a manually curated database of 2M bioactive molecules with known experimental data, especially with the inhibitory and binding properties against macromolecule targets. This database contains not only drug and drug-like molecules, but also natural products and biopolymers.

(4) **GDB-17** (Ruddigkeit et al., 2012) enumerates chemical structures that contain up to 17 atoms of C, N, O, S, and halogens, overlapping with the molecular weight range typical for lead

---

[5]ZINC-250k: `https://www.kaggle.com/datasets/1379f2461e75ef7a11d0c5ff3dd0c2440a6bcf33531bc0d303e8eac81a3a4b17`.

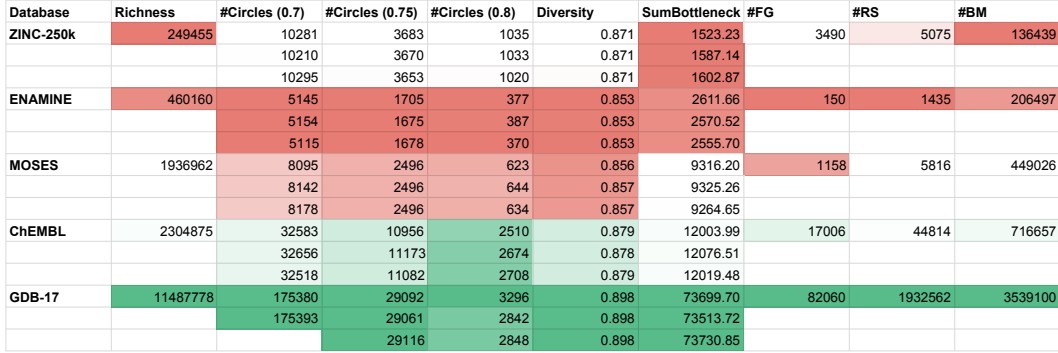

| Database | Richness | #Circles (0.7) | #Circles (0.75) | #Circles (0.8) | Diversity | SumBottleneck | #FG | #RS | #BM |
|---|---|---|---|---|---|---|---|---|---|
| ZINC-250k | 249455 | 10281 | 3683 | 1035 | 0.871 | 1523.23 | 3490 | 5075 | 136439 |
| | | 10210 | 3670 | 1033 | 0.871 | 1587.14 | | | |
| | | 10295 | 3653 | 1020 | 0.871 | 1602.87 | | | |
| ENAMINE | 460160 | 5145 | 1705 | 377 | 0.853 | 2611.66 | 150 | 1435 | 206497 |
| | | 5154 | 1675 | 387 | 0.853 | 2570.52 | | | |
| | | 5115 | 1678 | 370 | 0.853 | 2555.70 | | | |
| MOSES | 1936962 | 8095 | 2496 | 623 | 0.856 | 9316.20 | 1158 | 5816 | 449026 |
| | | 8142 | 2496 | 644 | 0.857 | 9325.26 | | | |
| | | 8178 | 2496 | 634 | 0.857 | 9264.65 | | | |
| ChEMBL | 2304875 | 32583 | 10956 | 2510 | 0.879 | 12003.99 | 17006 | 44814 | 716657 |
| | | 32656 | 11173 | 2674 | 0.878 | 12076.51 | | | |
| | | 32518 | 11082 | 2708 | 0.879 | 12019.48 | | | |
| GDB-17 | 11487778 | 175380 | 29092 | 3296 | 0.898 | 73699.70 | 82060 | 1932562 | 3539100 |
| | | 175393 | 29061 | 2842 | 0.898 | 73513.72 | | | |
| | | | 29116 | 2848 | 0.898 | 73730.85 | | | |

Figure 13: Measuring results on original databases.

| Database | Richness | #Circles (0.75) | Diversity | SumBottleneck | #FG | #RS | #BM |
|---|---|---|---|---|---|---|---|
| ZINC-250k | 173731 | 2035 | 0.861 | 1106.10 | 1952 | 2940 | 92715 |
| | | 2084 | 0.861 | 1096.43 | | | |
| | | 2040 | 0.863 | 1115.64 | | | |
| ENAMINE | 427366 | 1229 | 0.851 | 2461.75 | 142 | 1268 | 187742 |
| | | 1275 | 0.851 | 2402.39 | | | |
| | | 1263 | 0.851 | 2375.03 | | | |
| MOSES | 1846492 | 1714 | 0.856 | 8832.55 | 938 | 4513 | 413521 |
| | | 1698 | 0.855 | 8955.37 | | | |
| | | 1720 | 0.855 | 8971.22 | | | |
| ChEMBL | 967322 | 4570 | 0.872 | 5423.73 | 6304 | 14004 | 317255 |
| | | 4631 | 0.872 | 5464.92 | | | |
| | | 4543 | 0.873 | 5368.94 | | | |
| GDB-17 | 1621490 | 8094 | 0.889 | 10733.83 | 5969 | 167795 | 425582 |
| | | 8136 | 0.889 | 10670.31 | | | |
| | | 8114 | 0.889 | 10770.83 | | | |

Figure 14: Measuring results on filtered databases.

compounds. The lead-like subset filters 11M compounds with lead-like properties (100-350 MW & 1-3 clogP).

(5) Enamine is a company that provides compound libraries for high-throughput screening (HTS). The **Enamine Hit Locator Library**[6] is their largest diversity library with high MedChem tractability, and the compounds in the library are readily available for purchase and are guaranteed synthesizable at a reasonable cost.

## F.2 RESULTS ON CHEMICAL SPACE MEASURES

The measuring results are listed in Figure 13 and 14.

## G MOLECULAR GENERATION EXPERIMENT DETAILS

### G.1 MODEL IMPLEMENTATION

We implement the models with the official repositories[7]. All hyperparameters are set as default.

---

[6]ENAMINE diversity libraries: https://enamine.net/compound-libraries/diversity-libraries.

[7]RationaleRL: https://github.com/wengong-jin/multiobj-rationale.
DST: https://github.com/futianfan/DST.
JANUS: https://github.com/aspuru-guzik-group/JANUS.
MARS: https://github.com/bytedance/markov-molecular-sampling.

## G.2 MARS Variants

The terms incorporated into objectives are computed as follows[8]:

$$\text{Novelty}_{\text{Diversity}}(x, \mathcal{S}) := \frac{1}{|\mathcal{S}|} \sum_{y \in \mathcal{S}} d(x, y) \tag{11}$$

$$\text{Novelty}_{\text{SumBottleneck}}(x, \mathcal{S}) := \min_{y \in \mathcal{S}} d(x, y) \tag{12}$$

$$\text{Novelty}_{\#\text{Circles}}(x, \mathcal{S}) := \left[ \min_{y \in \mathcal{S}} d(x, y) > t \right] \tag{13}$$

For the model variants, we test different $\alpha$ values from $\{0.1, 0.3, 1.0, 3.0\}$ and report the best performance. Specifically, we use $\alpha = 1.0$ for MARS+Diversity, $\alpha = 0.3$ for MARS+SumBottlenck, and $\alpha = 0.1$ for MARS+#Circles. The threshold we use for the #Circles measure is $t = 0.60$.

## G.3 Results on Other Measures

Table 5: Measuring results of the chemical space explored by molecular generation methods. In each chemical space measure, the larger value the better. **Bold** indicates the best performance in each measure.

| Method | SumBottleneck | #FG | #RS | #BM |
|--------|---------------|-----|-----|-----|
| **Databases** | 280 | 58 | 132 | 502 |
| **RationaleRL** | $9 \pm 0.3\%$ | $39 \pm 0.0\%$ | $\mathbf{207 \pm 0.0\%}$ | $442 \pm 2.3\%$ |
| **DST** | 8 | 5 | 9 | 26 |
| **JANUS** | $76 \pm$ | $93 \pm$ | $73 \pm$ | $133 \pm$ |
| **MARS** | $535 \pm 7.9\%$ | $346 \pm 16.0\%$ | $67 \pm 3.5\%$ | $19.5\text{K} \pm 23.1\%$ |
| +Diversity | $715 \pm 12.9\%$ | $463 \pm 17.2\%$ | $67 \pm 1.9\%$ | $\mathbf{20.1K \pm 25.2\%}$ |
| +SumBot | $\mathbf{926 \pm 12.5\%}$ | $\mathbf{868 \pm 15.3\%}$ | $66 \pm 2.3\%$ | $14.0\text{K} \pm 20.8\%$ |
| +#Circles | $742 \pm 12.9\%$ | $601 \pm 35.0\%$ | $69 \pm 5.5\%$ | $18.9\text{K} \pm 15.3\%$ |

The results on other chemical space measures are listed in Table 5.

## G.4 Molecular Property and Binding Affinity Distributions

To investigate how incorporating chemical space measures into the objective as Eq. 11-13 can influence the optimization for molecular properties, we examine the property distributions of molecules generated by MARS and its variants.

As shown in Figure 15-16, our resulting libraries from the joint optimizations with chemical space measures (*i.e.*, MARS+#Circles and MARS+Diversity) are not showing a significant downward shift on JNK3 binding affinity as well as QED scores and SA scores. It is expected that compared with vanilla MARS, the property scores of molecules generated by MARS variants would decrease slightly, because instead of solely optimizing the property scores, the model needs to make sacrifices for a more diverse library. However, this sacrifice would be insignificant compared to the diversity gain in the resulting library. In practice, these molecules would pass the scoring threshold and would be considered qualified in these regards. In Table 2, the experiment results show that by incorporating chemical space measures into optimization objectives, the richness of discovered qualified molecules (JNK3 $\geq 0.5$, QED $\geq 0.6$, and SA $\leq 4$) got significantly improved.

## G.5 Visualization

To provide a more intuitive view of the effect that adding a joint objective of exploration can encourage the molecular generation model to discover a wider space, we visualize the chemical space explored by the baseline (MARS) and MARS+SumBottleneck as well as MARS+#Circles in Figure 17, where

---

[8]Equations 11-13 are approximations of $[\mu(\mathcal{S} \cup \{x\}) - \mu(\mathcal{S})]$ defined to avoid numerical issues and for computational efficiency.

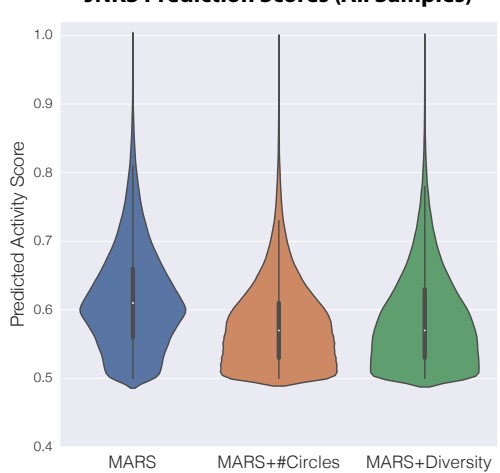 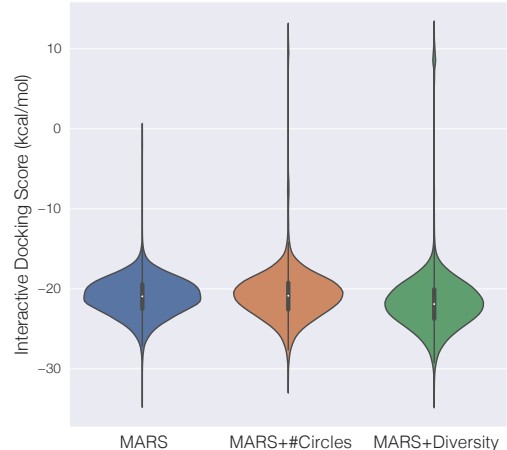

Figure 15: **Left**: Distributions of JNK3 machine learning model predicted score. We observe a downward trend in the MARS variant models, which are jointly optimized with a chemical space measure. However, all generated molecules are above the score threshold, and in practice, due to the limitation of the prediction model, they should be considered equally qualified candidates for further analysis. **Right**: Distributions of interaction scores of the ligands and the receptor docking (lower energy score indicates more favorable interactions). For the docking experiments, we randomly sampled 30K molecules from each generated library and performed molecular docking on the JNK3 target (PDB: 7KSI) using rDock (Ruiz-Carmona et al., 2014). The distributions of docking scores show that the joint optimizations on chemical space measures do not affect the binding affinity significantly when predicted by *in silico* docking.

we show the 2D layout of the Morgan fingerprints of the functional groups in the molecules generated through principal component analysis (PCA). The larger number of unique functional groups and the wider spread of the data points obtained by adding exploration-based novelty terms indicate more diverse structures being explored.

We show the molecular clustering results in Figure 18. Clusters are calculated based on Morgan fingerprints of the generated molecules and their Tanimoto similarity. Compared to the baseline model, a larger number of clusters can be obtained from MARS+SumBottleneck.

Figure 19 shows the dynamics of measures for MARS and its variants.

## H   FAST APPROXIMATION OF #CIRCLES

The computation of #Circles is combinatorial and the running time is exponential in theory. We implement two fast approximations by sacrificing accuracy as Algorithm 3 and Algorithm 4.

---

**Algorithm 3** Approximation of #Circles (sequential).

---

**Input:** The molecular set $\mathcal{S}$ to be measured, the distance metric $d$, the distance threshold $t$.
Randomly reorder the molecular set $\mathcal{S}$.
Initialize an empty set $\mathcal{C}$.
Set $n := |\mathcal{S}|$.
**for** $i$ **in** $\{1, \ldots, n\}$ **do**
    Add $\mathcal{S}_i$ to $\mathcal{C}$ if $\min_{y \in \mathcal{C}} \ d(x, y) > t$.
**end for**
**Return:** $|\mathcal{C}|$ and $\mathcal{C}$.

---

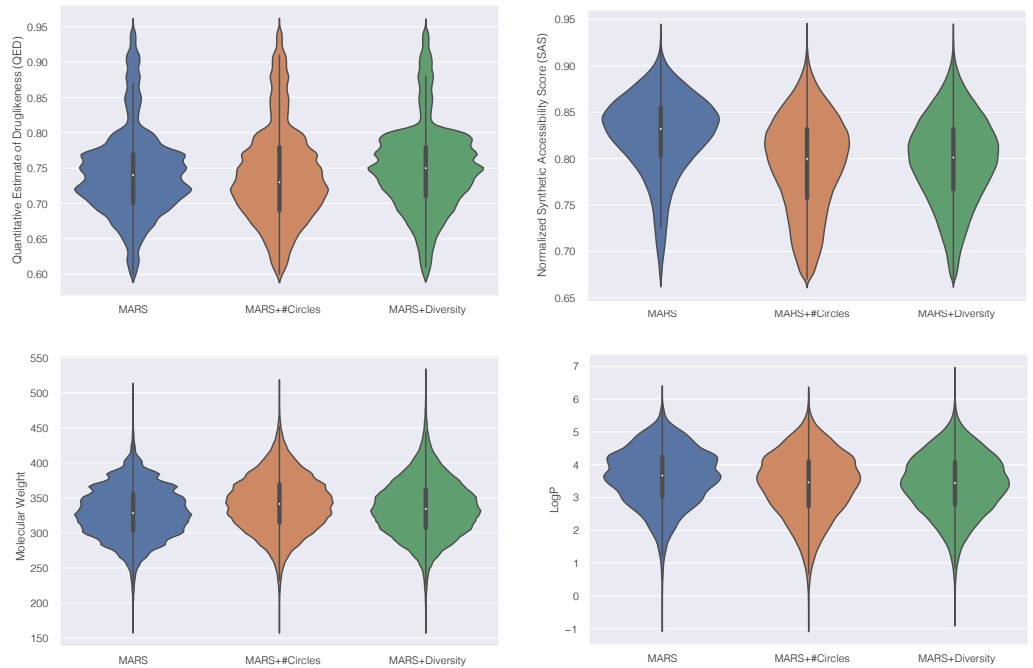

Figure 16: Selected molecular property distributions. Except for SAS (normalized from 0 to 1, the larger the better), all property distributions do not show significant changes across different models. The downward shift of the SAS is likely due to the increased diversity in the molecular libraries.

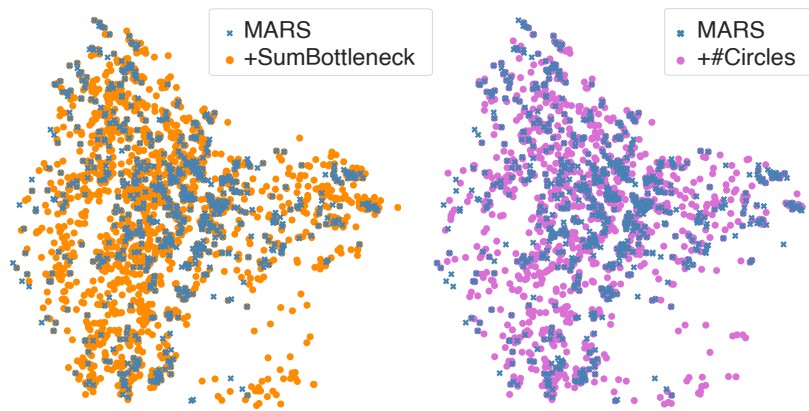

Figure 17: Optimizing chemical space measures encourages the molecular generation model to explore a larger span of the chemical space. This figure shows principal component analysis (PCA) of functional groups discovered by different models.

---

**Algorithm 4** Approximation of #Circles (recursive).

---

**Input:** The molecular set $\mathcal{S}$ to be measured, the distance metric $d$, the distance threshold $t$, maximum number of recursive layers $L$, number of processors $m$.
Call Algorithm 3 with arguments $(\mathcal{S}, d, t)$ if $L = 0$.
Randomly reorder the molecular set $\mathcal{S}$.
Evenly split $\mathcal{S}$ into $m$ subsets $\mathcal{S}_1, \dots, \mathcal{S}_m$.
Call Algorithm 4 with arguments $(\mathcal{S}_i, d, t, L-1, m)$ for $i = 1, \dots, m$ and collect the returns $\mathcal{C}_i$.
Set $\mathcal{C} := \mathcal{C}_1 \cup \cdots \cup \mathcal{C}_m$.
**Return:** Results obtained by calling Algorithm 3 with arguments $(\mathcal{C}, d, t)$.

---

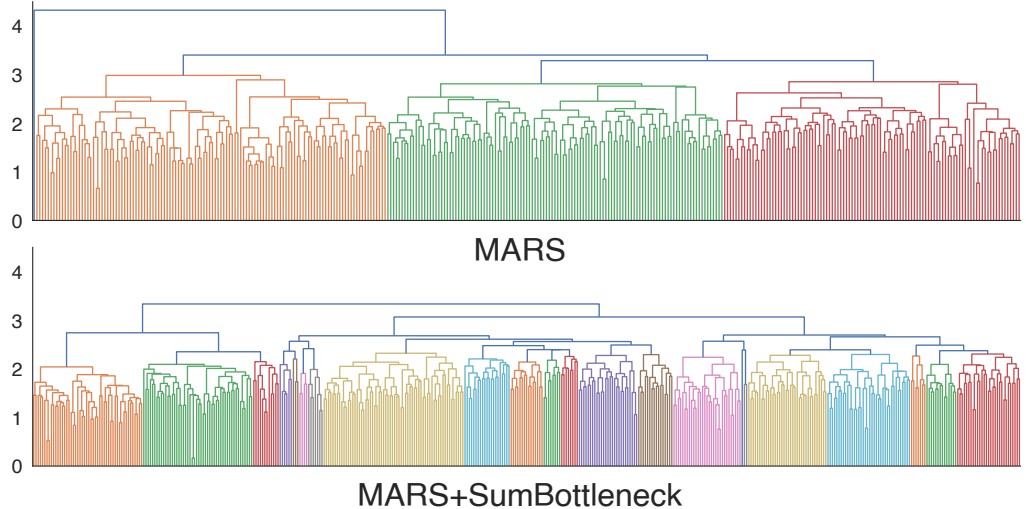

Figure 18: Hierarchical clustering results of generated molecules. Different colors stand for different clusters. The clustering threshold is set as $0.7\times$ the height of the highest linkage in accordance with SciPy's default value.

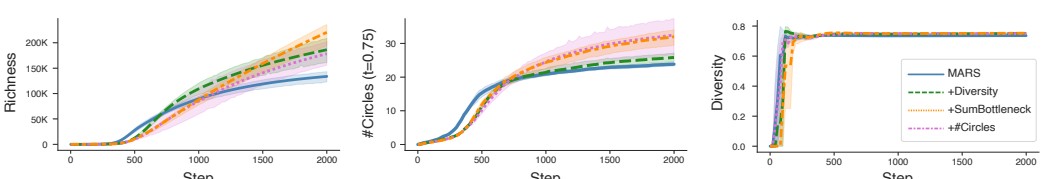

Figure 19: The dynamics of chemical space measures over generated molecules. Incorporating chemical measures into generation objectives increases the exploration of the chemical space.

