# OpenReview forum: "How Much Space Has Been Explored? Measuring the Chemical Space Covered by Databases and Machine-Generated Molecules"
_ICLR.cc/2023/Conference — ICLR 2023 poster_

### Official Review · Reviewer_74LY · 2022-10-17

**Confidence:** 4
**Correctness:** 3
**Technical Novelty And Significance:** 3
**Empirical Novelty And Significance:** 3
**Recommendation:** 6

**Clarity, Quality, Novelty And Reproducibility:**

Overall, the paper is well-written and provides a clear description of the problem that the authors are trying to address. In terms of novelty, some of the analyses provided I believe deserve credit:

* A deep look into metrics that measure coverage of chemical space
* An axiomatic analysis for the aforementioned metrics, with the proposal of two that intuitively most metrics should fulfill.
* A criteria to evaluate coverage metrics based on their correlation with biological functionality.
* Recommendations on which widely-known chemical databases cover the most space according to the metrics defined
* Insights into how large the coverage of molecules generated by de novo design algorithms


Sadly, none of the current study is reproducible as the authors claim that code will be released in the future. I believe that having an implementation of the provided metric (#circles) could already benefit other ongoing molecular de novo design studies.

**Strength And Weaknesses:**

Overall this is a good contribution to the field, w

Strengths:

* Provides new insight on the information that commonly-used metrics for de novo design provide.
* Formal, thorough analyses.
* Easy to read manuscript, provides enough context to understand the topic.
* Proposes new metric to evaluate chemical libraries addressing limitations of the existing ones

Weaknesses:

* Motivation for the presented axioms somewhat absent
* No accompanying code provided for the #circles metric, at least at time of submission.
* Exponential compute time for the proposed metric. Approximation not provided.

**Summary Of The Paper:**

In this paper the authors propose an axiomatic analysis of some of the metrics most commonly used in ML-based de novo design algorithms. Specifically, they lay down several axioms that measures of coverage of chemical space should obey and based on those propose a new metric for evaluation (#circles).

**Summary Of The Review:**

In general, I believe this to be a positive contribution to the field and I think the work provided here is novel enough to recommend publication. However, there are a few things/questions that the authors should address:

* In Figure 2, the #circles metric seems to outperform many of the other proposed metrics. Richness, in particular, as the authors mention, is a specific case of the #circles metric when t=0, yet performance-wise its correlation is even negative. Could the authors elaborate why they think this happens?
* On the threshold t=0.75. The authors claim that this choice is good for both the fixed size and the dynamically growing set settings - however this seems to have only been tested on the first setting (Fig 9).
* On section 5.2.3 the authors claim that ML models fail to explore a larger effectual area compared to databases. This conclusion is drawn after setting up a large baseline database, which will obviously cover a larger portion of chemical space than the generative methods, since the space of JNK3 binders with predefined characteristics should intuitively be smaller. I personally do not see this as a failure of the ML models, but I would love to have to see this further discussed in the manuscript. Along those lines, instead of using all 5 databases as a baseline, have the authors considered using a subset of these data sources where only JNK3 inhibitors are included?
* Another interesting insight is that the authors claim that the #circles metric can be tuned to explore either global or local regions of chemical space depending on which threshold t is chosen. I believe further attention should be paid to this, as it is only briefly mentioned in the results section. Along those lines, instead of choosing one threshold t, have the authors considered using a weighted average of the metric as a function of t?


Other minor points:

* Typo in section 4.2.2 (space before comma, after "is critical")
* In table 1, should the "axiomatic properties" header not point to Sec 4.1, instead of 3?
* Figure 9 (appendix) could use some rewriting
* Missing comma after Eq. 2
* Figure 4 is somewhat hard to read - consider increasing its size.

---

> ### Author Response · Authors · 2022-11-17
> **Response to Reviewer 74LY's comments**
>
> We appreciate the reviewer acknowledging the novelty of this study, which is encouraging. We also thank the reviewer for the constructive feedback and we address them in detail below.
>
> > **Q1: Motivation for the presented axioms somewhat absent.**
>
> Thank you for the comments and we agree that this part of the writing could be improved. For the subadditivity axiom, the motivation behind it is that, including more molecules should not decrease the degree of chemical space coverage. As it is easy to filter large candidate libraries into smaller subsets, including more molecules (even similar ones) is not harmful, and can render a higher probability of containing drug hits.
>
> For the dissimilarity axiom, molecular sets with more dissimilar molecules should have a higher degree of chemical space coverage. In the revised submission, we have reformulated the axiom for dissimilarity into a more intuitive form that encourages dissimilarity. Please refer to Axiom 4.2 in Sec. 4.1 in our revised submission for more details. We also re-examined the dissimilarity property for all example measures under the new axiom (see Appendix B). The results are mostly consistent except that the SumDiameter measure can satisfy the dissimilarity constraint now, although it still doesn’t satisfy the subadditivity constraint (Fig. 1).
>
> We would also like to stress that even though we have reformulated one of our axioms, the main conclusions and contributions of this paper remain unchanged.
>
> > **Q2: The performance of Richness.** In Figure 2, the #circles metric seems to outperform many of the other proposed metrics. Richness, in particular, as the authors mention, is a specific case of the #circles metric when $t=0$, yet performance-wise its correlation is even negative. Could the authors elaborate why they think this happens?
>
> Thanks for the question. We first note that in the experiments shown in Figure 2, as we are drawing molecules from a database, the Richness metric almost always equals the number of molecule draws, $n$. It is sometimes slightly smaller than n as some molecules in the database have the same SMILES representation so they are treated as duplicates. This explains why the correlation between Richness and the gold standard degenerates to be near zero and even negative in the *fixed-size* setting -- the Richness is fluctuating around the constant $n$ in this setting.
>
> > **Q3: On the threshold $t=0.75$.** The authors claim that this choice is good for both the fixed size and the dynamically growing set settings - however this seems to have only been tested on the first setting (Fig 9).
>
> We also tested the sensitivity of $t$ in the second setting. The results are displayed in Fig. 11, where we can see that $t=0.75$ is nearly optimal.
>
> > **Q4: On section 5.2.3 the authors claim that ML models fail to explore a larger effectual area compared to databases.** This conclusion is drawn after setting up a large baseline database, which will obviously cover a larger portion of chemical space than the generative methods, since the space of JNK3 binders with predefined characteristics should intuitively be smaller. I personally do not see this as a failure of the ML models, but I would love to have to see this further discussed in the manuscript. Along those lines, instead of using all 5 databases as a baseline, have the authors considered using a subset of these data sources where only JNK3 inhibitors are included?
>
> Thank you for suggesting this! Actually, we did filter the candidates from databases with the JNK3, QED, and SA scorers as described in Section 5.2.2. After filtering, only 1250 compounds satisfy all the constraints (i.e., $\text{JNK3}\ge0.5$, $\text{QED}\ge0.6$, and $\text{SA}\le4$), much less than the candidate set size 11M. The numbers in Table 2 all stand for positive candidates where three constraints are satisfied, thus all comparable. Unfortunately, ML models fail to surpass existing databases in this relatively fair comparison. We have clarified our evaluation settings in Sec. 5.2.2 to avoid confusion.

---

> ### Author Response · Authors · 2022-11-17
> **Response to Reviewer 74LY's comments (cont.)**
>
> > **Q5: On the $t$ parameter of #Circles.** Another interesting insight is that the authors claim that the #circles metric can be tuned to explore either global or local regions of chemical space depending on which threshold $t$ is chosen. I believe further attention should be paid to this, as it is only briefly mentioned in the results section. Along those lines, instead of choosing one threshold t, have the authors considered using a weighted average of the metric as a function of t?
>
> Thank you for the great suggestion! We also agree that we cannot focus on only one single $t$ value when evaluating and comparing databases as well as generative models with the #Circle measure, since different $t$ values capture different regimes of coverage – smaller $t$ tends to reflect local exploitation while larger $t$ relates more to global exploration. However, it might be difficult to find a universal set of weights to aggregate the results from different $t$. Inspired by the reviewer's suggestion, we further propose another intuitive way to compare and characterize databases and models in the revision.  In particular, we investigated how #Circles varies according to $t$ in the molecular generation setting. The #Circles curves are displayed in Appendix E.6 (Fig. 19), which reflects the characteristics of various molecular generation methods.
>
> > **Q6: No accompanying code provided** for the #circles metric, at least at time of submission. Exponential compute time for the proposed metric. Approximation not provided.
>
> We have added some descriptions to our approximation in Appendix F in our revision and referred to it in our main text. Please also refer to this anonymous repository for the code implementation of all chemical space measures (including the fast approximation for computing #Circles): https://anonymous.4open.science/r/chem-measure-1357.
>
> > **Q7: Some minor points.**
>
> We thank the reviewer for pointing out the typos. We have addressed these issues in our revision.

---

### Official Review · Reviewer_cGEX · 2022-10-20

**Confidence:** 4
**Correctness:** 3
**Technical Novelty And Significance:** 4
**Empirical Novelty And Significance:** 4
**Recommendation:** 8

**Clarity, Quality, Novelty And Reproducibility:**

# Clarity
The paper is well-organized and well-written, and it is very comfortable to read through.

# Quality
The quality is satisfactory except that there is a minor issue to be resolved. The equation in Definition 3.2 is actually Tanimoto similarity, and Tanimoto distance should be 1 - Tanimoto similarity. I assume that this is a typo, but I would appreciate if the authors re-confirm their software implementation.

# Novelty
Discussion on the coverage metric in this domain is novel to me.

# Reproducibility
It seems there exists enough information to reproduce the results. I would appreciate if the authors publish the source code.

**Strength And Weaknesses:**

# Strengths
- The problem setting itself is good, because the validity of existing metrics has been less focused in the domain.
- The proposed metric is intuitive and looks reasonable.
- Analyses on both the existing databases and molecules discovered by generative models are insightful and will be a basis for enhancing generative models.

# Weaknesses
### Axioms are not very convincing to me
The sub-additivity axiom implies that adding more and more molecules to $\mathcal{S}$ does not hurt the score, which may be reasonable for some purposes but may not be reasonable for others. For example, some of the existing methods aim to construct a focused library where molecules in it are more likely to satisfy the requirements. In such a case, adding irrelevant molecules to the generated library may improve the proposed coverage score, but is not desirable considering the original purpose. This indicates that "axioms" (which, I suggest, should be replaced by "assumptions" or "requirements") of the metrics depend on how the metrics are used. Therefore, I would ask the authors to clarify the usages of the metrics and deduce requirements for those usages.

In fact, it seems that the authors assume the scenario where the (focused) library is an input to the virtual screening module; given that scenario, adding irrelevant molecules to $\mathcal{S}$ is ok. In contrast, some of existing generative models may wish to substitute not only the library but also the virtual screening, i.e., they wish to generate hits directly. The proposed metric is not suitable to measure the diversity of the generated hits.

In summary, I would like the authors to discuss the scenario they consider and relate the axioms to the scenario tightly. In addition, I would like the authors to discuss the limitation of the proposed axioms and metrics, i.e., it may not be applied to other scenarios including evaluation of generative models of hits.

Regarding Axiom 4.2, what I don't understand is whether there always exists $x^\star$ such that $d(x^\star, x_1)=d(x^\star, x_2)=\frac{1}{2}d(x_1, x_2)$. If this statement does not hold, the axiom is not well-defined, and should be refined.



**Summary Of The Paper:**

The present paper is concerned about measuring how much a set of molecules covers the chemical space. All of the metrics that they consider take a set of molecules $\mathcal{S}$ as input and output a coverage metric. Among many existing heuristically-designed metrics, the authors propose a metric called #Circles, which amounts to the maximum number of non-overlapping circles of radius t/2 whose center are some of the molecules in $\mathcal{S}$.

The authors claim the validity of the proposed metrics in two ways. First, they propose two axioms that such a metric should satisfy, and conclude that the proposed metric satisfies both of them, while the others do not. Second, they compute the correlation between the coverage metric and a proxy gold standard of the variety of the molecules (which is the number of unique biological functionality types in $\mathcal{S}$), and show that the proposed metric correlates with the gold standard better than the others. Based on these two findings, the authors suggest that the proposed metric, #Circles, is preferred to the others.

The authors finally measure the coverage by the existing databases as well as the molecules generated by existing generative models. The analysis on the existing databases suggests that ZINC-250k and GDB-17 are recommended in terns of coverage. The analysis on generative models suggest that the sets of molecules discovered by existing ML-based methods have less coverage than the virtual screening.

**Summary Of The Review:**

This paper focuses on the validity of metrics used to evaluate the performance of generative models, and I like the problem setting. Although there are several issues to be resolved or to be further discussed in the paper, this paper could be a good starting point for us to understand and enhance evaluation metrics for molecular generation. Therefore, I would suggest to accept this paper, given the authors discuss the issues on the axioms in the paper (or if there is any misunderstanding, please correct me).

---

> ### Author Response · Authors · 2022-11-17
> **Response to Reviewer cGEX's comments**
>
> We really appreciate the encouraging comments and the detailed feedback. We address them in detail below.
>
> > **Q1: Axiom 4.1 is not very convincing.**
> > The sub-additivity axiom implies that adding more and more molecules to  does not hurt the score, which may be reasonable for some purposes but may not be reasonable for others. For example, some of the existing methods aim to construct a focused library where molecules in it are more likely to satisfy the requirements. In such a case, adding irrelevant molecules to the generated library may improve the proposed coverage score, but is not desirable considering the original purpose. This indicates that "axioms" (which, I suggest, should be replaced by "assumptions" or "requirements") of the metrics depend on how the metrics are used. Therefore, I would ask the authors to clarify the usages of the metrics and deduce requirements for those usages.
> > In fact, it seems that the authors assume the scenario where the (focused) library is an input to the virtual screening module; given that scenario, adding irrelevant molecules to  is ok. In contrast, some of existing generative models may wish to substitute not only the library but also the virtual screening, i.e., they wish to generate hits directly. The proposed metric is not suitable to measure the diversity of the generated hits.
> > In summary, I would like the authors to discuss the scenario they consider and relate the axioms to the scenario tightly. In addition, I would like the authors to discuss the limitation of the proposed axioms and metrics, i.e., it may not be applied to other scenarios including evaluation of generative models of hits.
>
> Thank you for the question.
>
> ## Suitable use cases
>
> We would like to clarify that the proposed metric can and often should be used together with other metrics concerning the effectiveness and safety of the drug candidates. We further elaborate this point through examples on how the proposed metric is used for model evaluation and model training in our paper.
>
> ### When using the proposed metric as an evaluation metric
>
> For example, the experiment results in Table 2 are obtained by the following procedure.
>
> - Step 1: Collect all the molecules generated by the models (or stored in a database);
> - Step 2: Filter the collected molecules based on effectiveness and safety metrics (e.g., $JNK3 \ge 0.5$);
> - Step 3: Calculate the coverage metrics using the **filtered** molecules.
>
> In this procedure, we are evaluating the coverage/diversity of the high-score molecules generated by the models. This is of practical interest because, in reality, the model-generated molecules will be filtered (like Step 2) and examined by downstream web experiments. A filtered set of molecules with higher coverage may lead to a better chance of hit.
>
> ### When using the proposed metric for model training
>
> We have also used the proposed metric to improve the model training (see MARS + #Circles in Table 2). In this case, the use of the proposed metric can be viewed as a regularization term in addition to the original effectiveness and safety objectives used in MARS.
>
> ## Evaluating generated hits
>
> We agree that, if there is a model that is able to directly generate drug hits with high precision, there is perhaps no need to encourage coverage/diversity in the first place.
>
> However, as mentioned in our Introduction, there is usually a considerable misalignmentnt between the easy-to-compute in silico property scores and the in vivo behaviors of the molecules. As a result, it may be very difficult to obtain a model that can generate durg hits with high precision in the near future, at least by merely optimizing the easy-to-compute property scores.
>
> It is the scenario where **many high-score molecules are NOT drug hits** that motivates the need for encouraging coverage/diversity of high-score molecules.
>
> ## Relating the axiom to the use cases
>
> As suggested by the two use cases mentioned earlier, for practical usage, we often consider the coverage of a set of molecules that already have good enough property scores. In such cases, we are always adding relevant molecules so it makes sense that the score does not hurt. The proposed metric serves as a complementary lens compared to the conventional property scores.

---

> ### Author Response · Authors · 2022-11-17
> **Response to Reviewer cGEX's comments (cont.)**
>
> > **Q2: Regarding Axiom 4.2**, whether there always exists $x^*$ such that $d(x^*,x_1*)=d(x^*,x_2)=\frac{1}{2}d(x_1,x_2)$. If this statement does not hold, the axiom is not well-defined, and should be refined.
>
> Thank you for pointing this out! Yes, the middle point $x^*$ does not always exist, and what we formulated in Axiom 4.2 is an ideal situation.
>
> We have reformulated this axiom for dissimilarity into a more intuitive form that encourages dissimilarity. Please refer to Axiom 4.2 in Sec. 4.1 in our revised submission for more details. We also re-examined the dissimilarity property for all example measures under the new axiom (see Appendix B). The results are mostly consistent except that the SumDiameter measure can satisfy the dissimilarity constraint now, although it still doesn’t satisfy the subadditivity constraint (Fig. 1).
>
> We would also like to stress that even though we have reformulated one of our axioms, the main conclusions and contributions of this paper remain unchanged.
>
> > **Q3: The equation in Definition 3.2 is actually Tanimoto similarity**, and Tanimoto distance should be 1 - Tanimoto similarity. I assume that this is a typo, but I would appreciate if the authors re-confirm their software implementation.
>
> Thank you for pointing this out! Yes, this is a typo in our draft. We have revised the equation and re-confirmed that the equation we used in our software implementation was correct.

---

### Official Review · Reviewer_UhAH · 2022-10-23

**Confidence:** 4
**Correctness:** 3
**Technical Novelty And Significance:** 4
**Empirical Novelty And Significance:** 3
**Recommendation:** 6

**Clarity, Quality, Novelty And Reproducibility:**

**Clarity**: the writing of this paper is generally good. My main recommendation would be for the authors to slightly change their terminology (e.g. change the word "measure", see above for more)

**Novelty/Originality**: from what I understand, the #Circles diversity measure and the classification of previously-used diversity measures is original and in my opinion highly valuable. I think that many people in ML for molecules would benefit from reading this paper (I personally benefitted from reading it).

**Quality**: There are good parts and bad parts. I think the #Circles diversity and the criticism of existing diversity measures is a very high-quality contribution. I think that other papers should immediately start using #Circles. However, to me the bad parts of the paper are the axioms, which are not super well-justified. I think the only indisputable axiom is monotonicity; the authors should either justify the rest or provide a more nuanced analysis which does not require the reader to agree with the strong forms of both axioms.

**Reproducibility**: code is not provided, and some key details are missing (e.g. the fast approximation to #Circles). I would consider this work not very reproducible.

**Strength And Weaknesses:**

**Strengths**

- _Important insight about previous diversity measures_: specifically, the proof/demonstration that the average pairwise distance between molecules in a set is not sub-additive is very important, because many previous papers published at ML conferences have used this measure. I think this paper makes a good case that this measure is fundamentally flawed and should not be used, which is itself an important contribution.
- _#Circles diversity measure is principled and useful_: it is interpretable, satisfies the proposed axioms, and intuitively makes a lot of sense. I think it can be used as a drop-in replacement for the flawed diversity measure above.
- _Classification of previously-used diversity measures is a nice insight_: I think that assembling and classifying many previously proposed methods to measure diversity is itself a useful contribution for people who work on molecules. While I am not certain about the novelty of this, I think many members of the ML community would benefit from reading it.
- _Implications beyond molecules_: although this paper focuses just on molecules, the findings about diversity measures and the proposal of the #Circles metric are completely general, and could be used to measure the diversity of data from other domains (e.g. images, point clouds, etc). I think the authors should state this more explicitly.

**Weaknesses**

- _Many un-numbered equations_: this made writing this review difficult and will make it difficult for people referring to parts of your paper to point to specific equations. Please give all equations numbers.
- _Questionable assumption of distance metric in section 3.1_: throughout the paper, the authors assume there is a reference distance metric $d$ on chemical space. I think this assumption is problematic: in reality there are _many_ possible distance metrics on chemical space, for example Tanimoto distances with many different types of fingerprints, descriptor-based distances (e.g. Euclidean distance between `rdkit` descriptors), or optimal-transport type distances. It is not really clear that any one metric is better than the others. What makes this problematic in my opinion is that the authors assume a single canonical metric $d$ and make this central to their analysis: for example in axiom 4.2. Clearly many diversity measures which satisfy axiom 4.2 for a given distance metric $d_1$ would no longer satisfy it if a new metric $d_2$ was used (but $d_1$ was still used internally by the diversity measure). I think it would be appropriate for the authors to take a more agnostic stance on this (e.g. simply saying that a metric $d$ can be used to define diversity, but it can also be defined in other ways).
- _Various confusing terminology/errors_.
  - The equation for Tanimoto distance in definition 3.2 actually seems to define Tanimoto similarity (i.e. $d(x,x)=1$ instead of $d(x, x)=0$. The authors probably meant to use $1-d$ as the definition.
  - The term "measure" in definition 3.3 is a poorly chosen name in my opinion because "measure" is already a loaded term from measure theory describing a similar function which also satisfies other properties (notably sub-additivity). Given that the authors intend to define a measure here as a function which is _not_ necessarily sub-additive, I think this may confuse many readers. I would suggest instead "diversity measure" or "diversity metric" to clarify this.
  - The term "similarity matrix" is used on page 4 without being defined in general. For the Tanimoto metric there is an obvious notion of similarity ($1-d$), but for other distance metrics I don't think this is the case.
  - "Cover" is not really defined in section 3.2.2: it is unclear what the range of a "cover" function is, not what its properties should be. I would instead try to define this using a suitable distance metric (e.g. Tanimoto distance with fingerprints which explicitly encode the presence of scaffolds). Coverage could then be restated in a similar way to circles, e.g. $\mathrm{Coverage}(\mathcal{S}, \mathcal{R}) = \sum_{y\in\mathcal{R}} \mathbb{I}\left[\exists x\in\mathcal{S}: d(x, y) < t\right]$. This is counting the number of molecules in the reference set with a sufficiently similar molecule in $\mathcal{S}$
  - #Circles in equation 1 will always equal 0. I think the authors need to add the additional condition $x\neq y$ to get the desired behaviour.
- _Small issues with classification system in section 3.2_: I don't understand why #Circles is not a distance-based measure: it also uses a distance metric, and is essentially using the matrix of all pairwise distances between molecules like most of the methods in section 3.2.1. I think that putting it in its own section is a bit confusing, and downplays the connection between #Circles and all the other diversity measures. Second, I think that the parameters of many of these metrics should be more clearly stated.
  - I would give a subscript $d$ to all measures in section 3.2.1 (e.g. $\mathrm{Diversity}_d, $ ) to clearly emphasize the dependence on a chosen distance metric $d$.
  - Similarly, I think you should write $\mathrm{Circles}_{d,t}$
- _Axioms in section 4 are not well-justified_:
  - Axiom 4.1 is justified by saying that increasing the size of the set should not decrease the diversity.  First, while I believe this, I don't know that everybody would believe this intuitively. I think you need to argue more explicitly that diversity should be measuring "coverage" and not "average similarity". If I were to write the argument, I would essentially say "a large set of molecules can be easily filtered to a smaller set so having many similar molecules is not a bad thing: what really matters is the region of chemical space explored by the list". Second, this intuitive justification only justifies the first inequality, not the second ($\mu(S_1\cup S_2) \leq \mu(S_1) + \mu(S_2)$). The reason for requiring this inequality is completely unclear to me. I know that a similar inequality appears in measure theory but I don't know why it applies here. I actually think it should be removed. This slightly weaker axiom is less disputable, and would still allow corollaries A.2 and A.3 to hold. It would slightly change the classification in Figure 1 (e.g. diameter would now be sub-additive). I would then also rename this axiom to "monotonicity".
  - Axiom 4.2 does not seem to match the description given: the description states that more dissimilar molecules should be given a higher diversity when added to a set, but the axiom seems to state that when trading off distance between 2 elements of a set and a third element, the trade-off with an even distance between the two points should be favoured. This seems sort of reasonable to me, but it is not at all clear that this should be required. I would certainly not describe it as "simple and intuitive". I think the authors need to provide better justification for this axiom or re-formulate it. Also, it depends on the distance metric used which I think is undesirable (c.f. my discussion above).
- _No details on how to compute circles diversity_: the authors state that computing#Circles is combinatorial with exponential running time, and that they use a fast approximation instead. However this approximation does not appear to be described anywhere. Given that #Circles is a key contribution of this paper, I think the authors need to explicitly state this, ideally in the main text instead of the appendix.

**Summary Of The Paper:**

This paper discusses techniques to measure the coverage and diversity of a subset of chemical space (i.e. a set of molecules). The authors give a mathematical definition of a diversity measure, then postulate two axioms which these measures should satisfy: sub-additivity and dissimilarity. This leads to their 3 main contributions:

1. Classifying 3 different families of diversity measures
2. Showing that many popular diversity measures do not satisfy their axioms
3. Proposing the #Circles measure which does satisfy both axioms.

The authors then perform several molecular screening and design experiments with the diversity measures discussed above, including the #Circles measure.

**Summary Of The Review:**

To me the good parts of the paper are the #Circles diversity and the criticism of other diversity measures, which I think the ML community needs to hear and digest to improve the quality of work in this area. However, numerous small issues make me think the paper is not quite ready to be accepted in its current form. Chief among these is that the axioms are not very well-justified in my opinion. This part of the paper is too important to be glossed over.

I really like the #Circles measure and would like to accept this paper, so I would propose the following changes for my score to increase:
1. Change axioms
2. Fix notation issues and errors which I pointed out above
3. Better discussion of dependence on $d,t$ for all metrics (these are somewhat arbitrary choices which will have a large influence on the results unfortunately)

---

> ### Author Response · Authors · 2022-11-17
> **Response to Reviewer UhAH's comments**
>
> We really appreciate the very detailed and constructive feedback. The reviewer's acknowledgment of the novelty of this study is very encouraging to us, and most of the critiques are fair. And we are happy to report that we are able to leverage most of the constructive feedback into our revised version.
>
> > **Q1: Questionable assumption of distance metric in section 3.1.**
> > Throughout the paper, the authors assume there is a reference distance metric $d$ on chemical space. I think this assumption is problematic: in reality there are many possible distance metrics on chemical space, for example Tanimoto distances with many different types of fingerprints, descriptor-based distances (e.g. Euclidean distance between rdkit descriptors), or optimal-transport type distances. It is not really clear that any one metric is better than the others. What makes this problematic in my opinion is that the authors assume a single canonical metric $d$ and make this central to their analysis: for example in axiom 4.2. Clearly many diversity measures which satisfy axiom 4.2 for a given distance metric $d_1$ would no longer satisfy it if a new metric $d_2$ was used (but $d_1$ was still used internally by the diversity measure). I think it would be appropriate for the authors to take a more agnostic stance on this (e.g. simply saying that a metric $d$ can be used to define diversity, but it can also be defined in other ways).
>
> We would like to clarify that our definition and analysis are not restricted to any particular distance definition, as long as this distance function obeys the four axioms of metric space.
>
> Our definition (Def. 3.1) and proofs (Appendix B) are general enough to accommodate all (but not limited to) the following distance functions: The Tanimoto distances with different types of fingerprints, Euclidean distances with different types of descriptors, and some optimal-transport type distances (e.g., the Wasserstein metric), are all metric functions. We have also provided other examples under Eq. 1.
>
> In our main paper, our empirical studies (Sec. 4.2) are all based on the Tanimoto distance. However, in Appendix C.2-3, we further had an empirical study based on a VAE-based distance, where we found similar empirical conclusions as the Tanimoto distance. In our revisions, we have stressed the generality of our definitions and its extensibility on different types of distance metrics (Before Def. 3.3).
>
> > **Q2: Axiom 4.1 is not well-justified.** Axiom 4.1 is justified by saying that increasing the size of the set should not decrease the diversity. First, while I believe this, I don't know that everybody would believe this intuitively. I think you need to argue more explicitly that diversity should be measuring "coverage" and not "average similarity". If I were to write the argument, I would essentially say "a large set of molecules can be easily filtered to a smaller set so having many similar molecules is not a bad thing: what really matters is the region of chemical space explored by the list". Second, this intuitive justification only justifies the first inequality, not the second ($\mu(\mathcal{S}_1\cup\mathcal{S}_2)\le\mu(\mathcal{S}_1)+\mu(\mathcal{S}_2)$). The reason for requiring this inequality is completely unclear to me. I know that a similar inequality appears in measure theory but I don't know why it applies here. I actually think it should be removed. This slightly weaker axiom is less disputable, and would still allow corollaries A.2 and A.3 to hold. It would slightly change the classification in Figure 1 (e.g. diameter would now be sub-additive). I would then also rename this axiom to "monotonicity".
>
> We appreciate the reviewer pointing out the gap in the understanding of Axiom 4.1. We have elaborated the motivation for presenting Axiom 4.1 in our revision according to the reviewer’s constructive suggestion (the first paragraph in Sec. 4.1 and the paragraph after Axiom 4.1).
>
> For the second inequality, we would like to further comment on why this is desired in the context of drug discovery. In empirical drug design, a molecular set that covers a wider space tends to have a higher probability of including a hit. In this sense, the second inequality is reasonable because the probability that the combined set contains a hit can not exceed the sum of hit probabilities of the two original sets.

---

> > ### Comment · Reviewer_UhAH · 2022-11-17
> > **Response to all reviewer comments**
> >
> > Thank you for your responses and updates to to the paper. I think the newer version is a good improvement on the original version. Let me respond to the points you made:
> >
> > - Q1 (distance metric): I appreciate the emphasis on generality; I am happy with these changes
> > - Q2 (axiom 4.1): I like the added justification for the first inequality. I think the case for the second inequality is much weaker than the first inequality, but I do like the motivation in terms of probabilities. _I think you should make this argument explicitly in the paper (you do not seem to make it in the current version), and consider splitting the second inequality into its own axiom._ Even though a 3-axiom version may make it hard to plot Figure 1, I think it would make the paper's contribution stronger. I think that Figure 1 is one of the best parts of the paper: it could serve as a handy reference for practitioners who are trying to choose a diversity metric. By lumping the first inequality (monotonicity) with the second (outer measure), I think you lose fine-grained information about the strengths/weaknesses about each metric. In my work, I feel like monotonicity is extremely important, but axiom 4.2 and the "outer measure" property are not important, and it would be nice to see from this figure which diversity measures are monotonic and which are not.
> > - Q3 (axiom 4.2): I like the revised presentation and justification.
> > - Q4 (dependence on t, d): I like these revisions
> > - Q5 (beyond molecules): good revision
> > - Q6 (terminology): good changes
> > - Q7 (renaming of the classes of metrics): good change, I like the term "aggregation-based"
> > - Q8/Q9: good changes
> > - Q10 (equation numbers): good change, although I think each diversity measure in equation 3 should have its own equation number (6 separate definitions with one number is a bit much, no?)
> >
> > Overall I am happy with these changes and will increase my score for now.

---

> > > ### Author Response · Authors · 2022-11-18
> > > **Response to Reviewer UhAH’s further comments**
> > >
> > > We thank the reviewer for the quick response and additional suggestions! We agree with the suggestions and have further revised our paper. In this revision, we made the following changes to our paper:
> > >
> > > 1. **Regarding Q2 (subadditivity)**, we have split our original subadditivity axiom into two independent axioms (i.e., monotonicity and subadditivity). We have also included the intuition of “hitting probability” in the main text of the current version. Please refer to Axiom 4.1-2 in Sec. 4.1. We also provided proofs for both monotonicity and subadditivity in Appendix B. We updated Fig. 1 accordingly. One thing we would like to mention is that currently we are not able to prove or disprove the subadditivity for DPP, and can only disprove the monotonicity for DPP. We will try to add the proof later in the camera-ready submission if the paper can be accepted by the conference.
> > > 2. **Regarding Q10 (equation numbering)**, we have added subequation labels for all measure definitions in Eq. 3.
> > >
> > > Thank you again for the valuable and constructive feedback! Hope the revision addresses your concerns.

---

> > > > ### Comment · Reviewer_UhAH · 2022-11-18
> > > > **Good changes**
> > > >
> > > > I think these changes make the paper much clearer and stronger. I look forward to seeing the next version of the manuscript with additional proofs :)

---

> > > > > ### Author Response · Authors · 2022-11-18
> > > > > **Thank the reviewer for the nice words!**
> > > > >
> > > > > Thanks for your nice words! We also think your suggestions have helped us make our contribution much clearer. It was a joy addressing your comments and we hope you consider the paper acceptable for the conference.

---

> ### Author Response · Authors · 2022-11-17
> **Response to Reviewer UhAH's comments (cont. 1)**
>
> > **Q3: Axiom 4.2 does not seem to match the description given**: the description states that more dissimilar molecules should be given a higher diversity when added to a set, but the axiom seems to state that when trading off distance between 2 elements of a set and a third element, the trade-off with an even distance between the two points should be favoured. This seems sort of reasonable to me, but it is not at all clear that this should be required. I would certainly not describe it as "simple and intuitive". I think the authors need to provide better justification for this axiom or re-formulate it. Also, it depends on the distance metric used which I think is undesirable (c.f. my discussion above).
>
> Thanks for the comments! We have followed the reviewer's suggestions and reformulated the axiom for dissimilarity into a more intuitive form that encourages dissimilarity. Please refer to Axiom 4.2 in Sec. 4.1 in our revised submission for more details. We also re-examined the dissimilarity property for all example measures under the new axiom (see Appendix B). The results are mostly consistent except that the SumDiameter measure can satisfy the dissimilarity constraint now, although it still doesn’t satisfy the subadditivity constraint (Fig. 1).
>
> We would also like to stress that even though we have reformulated one of our axioms, the main conclusions and contributions of this paper remain unchanged.
>
> For the dependence of the distance metric, please refer to the response to Q1 and Appendix B, where we have demonstrated that the proofs of Axiom 4.2 do not rely on the specific definition of the distance metric d. This is true for both our original formulation of Axiom 4.2 and the new formulation.
>
> > **Q4: Better discussion of dependence on $d$ and $t$  for all metrics (these are somewhat arbitrary choices which will have a large influence on the results unfortunately).**
>
> Thanks for your suggestions! For the dependence on $d$, we assume this question is a summary of Q1. In our response to Q1, we have clarified that the definition and proofs do not depend on the particular choice of metric function. In addition, we also have included empirical studies in Appendix C.2-3 for metric functions other than Tanimoto distance (i.e., a VAE-based distance).
>
> For the dependence on $t$, we actually have included a sensitivity test of the threshold $t$ in the original submission (Appendix C.2 and C.3). From the results (Fig. 8 and Fig. 11), we can see that the variation of #Circles with respect to $t$ is smooth in the range of $[0.40, 0.75]$ in the fixed-size setting, and $[0.40, 0.80]$ in the growing-size setting. Both settings agree that the optimal value of $t$ is around $0.75$.
>
> In addition, we want to stress that the parameter $t$ has an interpretable meaning. When $t$ is smaller, the corresponding #Circles measure tends to reflect the "spread" of molecules at a finer granularity. When $t$ is larger, the corresponding #Circles focuses more on the global layout of the molecule distribution. In the revised version, we also add further investigation on how #Circles varies according to $t$ in the molecular generation setting. The #Circles curves are displayed in Appendix E.6 (Fig. 19), which reflects the characteristics of various molecular generation methods.
>
> > **Q5: Implications beyond molecules**: although this paper focuses just on molecules, the findings about diversity measures and the proposal of the #Circles metric are completely general, and could be used to measure the diversity of data from other domains (e.g. images, point clouds, etc). I think the authors should state this more explicitly.
>
> Thank you for the suggestion! We totally agree that our evaluation framework on coverage/diversity measures as well as the proposed #Circles measure could also be applied to other domains like the science of science and material design. We listed the examples of image/text generation at the end of the Conclusion. We have further elaborated this in our revision (by the end of the Conclusion).

---

> ### Author Response · Authors · 2022-11-17
> **Response to Reviewer UhAH's comments (cont. 2)**
>
> > **Q6: Various confusing terminology/errors.**
>
> > **Q6.1: The equation for Tanimoto distance in definition 3.2 actually seems to define Tanimoto similarity.**
>
> Thank you for pointing this out! This is indeed a typo and we have revised it in the new version.
>
> > **Q6.2: The term "measure" in definition 3.3 is a poorly chosen name in my opinion** because "measure" is already a loaded term from measure theory describing a similar function which also satisfies other properties (notably sub-additivity). Given that the authors intend to define a measure here as a function which is not necessarily sub-additive, I think this may confuse many readers. I would suggest instead "diversity measure" or "diversity metric" to clarify this.
>
> Thank you for the suggestion! We actually defined our metrics as “chemical space measures” instead of “measures”. We chose the term “measure” because of two reasons.
>
> First, though Axiom 3.1 defies the strict requirement of countable additivity in measure theory, by satisfying subadditivity, our chemical space measure serves as a particular instance of outer measures. We removed the discussion on the connection to mathematical measures in our main paper due to the page limitation. We will add it back in a later revision if the page limitation permits. Please refer to our discussion in Appendix A.
>
> Second, though “diversity” is a very good term that can describe our intention of encouraging diverse candidate libraries, this term is already been used in the literature and often refers to “internal diversity”. Therefore, to distinguish our more generic definition to the particular internal diversity measure, we use “chemical space measures” to summarize such metrics in the chemical space.
>
> > **Q6.3: The term "similarity matrix" is used on page 4 without being defined in general.** For the Tanimoto metric there is an obvious notion of similarity ($1-d$), but for other distance metrics I don't think this is the case.
>
> Thank you for pointing this out! We have modified the definition of the DPP measure in our revised version. Now this measure is defined specifically with Tanimoto similarity.
>
> > **Q6.4: "Cover" is not really defined in section 3.2.2**: it is unclear what the range of a "cover" function is, not what its properties should be. I would instead try to define this using a suitable distance metric (e.g. Tanimoto distance with fingerprints which explicitly encode the presence of scaffolds). Coverage could then be restated in a similar way to circles, e.g. $\text{Coverage}(\mathcal{S},\mathcal{R})=\sum_{y\in\mathcal{R}}\mathbb{I}[\exists x\in\mathcal{S}: d(x,y)<t]$. This is counting the number of molecules in the reference set with a sufficiently similar molecule in $\mathcal{S}$.
>
> Thank you for the advice! Coincidentally, at the early stage of this work, we considered a definition similar to the one suggested by the reviewer. However, if adopting this definition, the widely used reference measures like #FG, #RS, and #BM will be ruled out, since these measures are not calculated based on distances. So we eventually left the "cover" function as an "abstract" function that can be "reloaded" for different measures.
>
> Nevertheless, we agree that the metric suggested by the reviewer is another good example of the possible designs of reference-based measures. We added a theoretical analysis of this metric by the end of Appendix B.
>
> > **Q6.5: #Circles in equation 1 will always equal 0. I think the authors need to add the additional condition to get the desired behaviour.**
>
> Thank you for pointing this out! We have changed the condition in Eq. 4 to $x\ne y$ in our revision.
>
> > **Q7: Small issues with classification system in section 3.2**: I don't understand why #Circles is not a distance-based measure: it also uses a distance metric, and is essentially using the matrix of all pairwise distances between molecules like most of the methods in section 3.2.2 (previously Sec. 3.2.1 in the original submission). I think that putting it in its own section is a bit confusing, and downplays the connection between #Circles and all the other diversity measures.**
>
> Thank you for your suggestions! It is a good point that the *distance-based* measures and the *locality-based* measures are both based on chemical space distances. The reason why we group measures in Sec. 3.2.2 together is that they are all metrics obtained by simply aggregating pairwise distances. We have changed the category name to *aggregation-based* in our revision and also added a discussion on the connection between *aggregation-based* measures and *locality-based* measures.

---

> ### Author Response · Authors · 2022-11-17
> **Response to Reviewer UhAH's comments (cont. 3)**
>
> > **Q8: The parameters of many chemical space measures should be more clearly stated.**
>
> Thank you for the suggestion! We have added $d$ and $t$ as the parameters in the definitions of measures.
>
> > **Q9: No details on how to compute circles diversity**: the authors state that computing #Circles is combinatorial with exponential running time, and that they use a fast approximation instead. However this approximation does not appear to be described anywhere.
>
> We have added some descriptions to our approximation in Appendix F in our revision and referred to it in our main text. Please also refer to this anonymous repository for the code implementation of all chemical space measures (including the fast approximation for computing #Circles): https://anonymous.4open.science/r/chem-measure-1357.
>
> > **Q10: Many un-numbered equations.** This made writing this review difficult and will make it difficult for people referring to parts of your paper to point to specific equations. Please give all equations numbers.
>
> Thank you for pointing this out! We have labeled the un-numbered equations in our revision.

---

### Official Review · Reviewer_yinU · 2022-10-25

**Confidence:** 4
**Correctness:** 3
**Technical Novelty And Significance:** 3
**Empirical Novelty And Significance:** 2
**Recommendation:** 6

**Clarity, Quality, Novelty And Reproducibility:**


The paper is clear written and it provides the reader a new perspective on this problem.
 The experiment results indicate that this metric is too sensitive to the parameter, which make is impractical to use this metric to evaluation molecule generation models.


**Details Of Ethics Concerns:**

No ethics concerns.

**Strength And Weaknesses:**

Strengths:

The authors provide a new perspective on the metrics for measuring the chemical space and conduct various experiments for molecule generation to demonstrate the argument.

Weaknesses:

 1) #Circles is  sensitive to the distance threshold parameter t, according to Table 2. MARS with #Circles is notably better than RationaleRL when t is 0.70; but theses two models achieve similar #Circles scores when t is 0.80. Such sensitivity to distance threshold parameter will make #Cricles impractical to evaluate model performance, i.e., the score can be controlled by selecting various distance threshold parameter to present inconsistent results.

 2) Is the setting of Axiom4.2(Dissimilarity) necessary in defining the measure of the topological space and what is the significance in the field of realistic drug design?

 3) Are there any relevant experiments that can demonstrate the formulation of Eq. 3 to generate higher quality molecules? What kind of model will the authors design for this purpose.

************************
Considering the reviewers' detailed response and revision, I have raised the score.



**Summary Of The Paper:**

The paper proposes a new metric called #Circles for chemical spacing measurement. The authors analyzed and compared existing metrics and their metric mathematically, proving that their metric has two good properties. They also conducted experiment to analyze current datasets and molecule generation models by their metric.

**Summary Of The Review:**

The paper provides enough mathematical proof to confirm their proposed metric has good theoretical properties. But the experimental results indicate that this metric is sensitive to the parameter, which make is impractical to use this metric to evaluation molecule generation models. And the model equipped with this metric did not show significant improvement over other metrics.

---

> ### Author Response · Authors · 2022-11-17
> **Response to Reviewer yinU's comments**
>
> We thank the reviewer for the constructive comments. The detailed responses are provided below.
>
> > **Q1: #Circles is very sensitive to the distance threshold parameter $t$.**
> > According to Table 2. MARS with #Circles is notably better than RationaleRL when $t$ is 0.70; but theses two models achieve similar #Circles scores when $t$ is 0.80. Such sensitivity to distance threshold parameter will make #Cricles impractical to evaluate model performance, i.e. the score can be controlled by selecting various distance threshold parameters to present inconsistent results. The experimental results indicate that this metric is sensitive to the parameter, which make is impractical to use this metric to evaluation molecule generation models.
>
> Thank you for the great question. Indeed the #Circles can vary with respect to the parameter $t$ and we actually have included a sensitivity test of the threshold $t$ in the original submission (Appendix C.2 and C.3). From the results (Fig. 8 and Fig. 11), we can see that the variation of #Circles with respect to $t$ is smooth in the range of $[0.40, 0.75]$ in the fixed-size setting, and $[0.40, 0.80]$ in the growing-size setting. Both settings agree that the optimal value of $t$ is around $0.75$.
>
> In addition, the parameter $t$ has an interpretable meaning. When $t$ is smaller, the corresponding #Circles measure tends to reflect the "spread" of molecules at a finer granularity. When $t$ is larger, the corresponding #Circles focuses more on the global layout of the molecule distribution.
>
> This intuition explains the results of RationaleRL and MARS in Table 2. RationaleRL tends to explore multiple global areas because it is designed to combine active fragments (rationales), while MARS is more capable of further exploiting high-quality local regions and generating many effective local variants due to its MCMC implementation. So MARS is better than RationaleRL when $t$ is smaller and the corresponding measure is zooming into finer granularity, while the two get closer when $t$ is larger. This finding also aligns with the observations in the study of [1], where the authors found RationaleRL tends to generate small clusters across the chemical space while MARS tends to heavily exploit the already discovered regions (see Fig. 3 (a),(c) of [1]).
>
> In the revised version, we also add further investigation on how #Circles varies according to $t$ in the molecular generation setting. The #Circles curves are displayed in Appendix E.6 (Fig. 19), which reflects the characteristics of various molecular generation methods.
>
> In summary, we would like to stress that the results in Table 2 cannot show the sensitivity of t, but rather indicators for the characteristics of molecular generation algorithms.
>
> [1] MARS: Markov Molecular Sampling for Multi-objective Drug Discovery, Xie et al., ICLR 2021
>
> > **Q2: Is the setting of Axiom4.2 (Dissimilarity) necessary in defining the measure of the topological space and what is the significance in the field of realistic drug design?**
>
> Yes, dissimilarity/diversity is indeed a critical consideration in practical drug design. As we suggested in the Introduction, only a tiny fraction of the high-score molecules can lead to drug hits. In reality, given a limited budget for the expensive downstream wet-lab experiments, having a diverse set of drug candidates can significantly increase the chance of drug hits.
>
> To better reflect our description and motivation for encouraging dissimilarity, we have reformulated this axiom into a more intuitive form. Please refer to Axiom 4.2 in Sec. 4.1 in our revised submission for more details. We also re-examined the dissimilarity property for all example measures under the new axiom (see Appendix B). The results are mostly consistent except that the SumDiameter measure can satisfy the dissimilarity constraint now, although it still doesn’t satisfy the subadditivity constraint (Fig. 1).
>
> We would also like to stress that even though we have reformulated one of our axioms, the main conclusions and contributions of this paper remain unchanged.
>
> [2] Rational methods for the selection of diverse screening compounds, Huggins et al., ACS Chemical Biology 2021
>
> [3] Toward performance-diverse small-molecule libraries for cell-based phenotypic screening using multiplexed high-dimensional profiling, Wawer et al., PNAS 2014

---

> > ### Comment · Reviewer_UhAH · 2022-11-17
> > **dependence on t does not make circles "impractical to evaluate model performance"**
> >
> > I disagree with reviewer yinU's claim that the dependence on t is a weakness of the proposed circles metric. The reviewer's claim that " the score can be controlled by selecting various distance threshold parameter to present inconsistent results" doesn't really make sense: if #Circles was used as a metric for a benchmark, obviously the same distance metric $d$ and distance threshold $t$ would need to be chosen to ensure that the same metric is being used.
> >
> > I think your concern might really be about the presentation of #Circles. My interpretation is that #Circles is a _family_ of diversity measures index by $d,t$ (i.e. each $d,t$ pair defines a unique metric). By not emphasizing this dependence in the #Circles on $d,t$ some people may [incorrectly] assume that $d,t$ can be freely chosen, but this would not be the case in a benchmark. I think it would be clearer if the authors wrote this dependence explicitly, e.g.
> > $\mathrm{Circles}_{d=Tanimoto,t=0.5}$.
> >
> > I think you should consider the _strength_ of this dependence on $d,t$: chiefly that this metric has a very intuitive interpretation, which many other diversity measures do not have. I encourage you to reconsider your position on this aspect of #Circles.

---

> ### Author Response · Authors · 2022-11-17
> **Response to Reviewer yinU's comments (cont.)**
>
> > **Q3: Are there any relevant experiments that can demonstrate the formulation of Eq. 9 (previously Eq. 3 before revision) to generate higher quality molecules? What kind of model will the authors design for this purpose.**
>
> We would like to answer this question in two different interpretations of the “higher quality molecules” here.
>
> When “higher quality” means the molecules exhibit high property scores (e.g., binding affinity, QED, and SA), please refer to Fig. 14-15 in Appendix E for the comparison between property score distributions obtained with MARS variants. Our resulting libraries from the joint optimizations with chemical space measures (*i.e.*, MARS+\#Circles and MARS+Diversity) didn’t show a significant downward shift in JNK3 binding affinity as well as QED scores and SA scores. It is expected that compared with vanilla MARS, the property scores of molecules generated by MARS variants would decrease slightly, because instead of solely optimizing the property scores, the model needs to make sacrifices for a more diverse library. However, this sacrifice would be insignificant compared to the diversity gain in the resulting library. In practice, these molecules would pass the scoring threshold and would be considered qualified in these regards. In Table 2, the experiment results show that by incorporating chemical space measures into optimization objectives, the richness of discovered qualified molecules ($\text{JNK3}\ge0.5$, $\text{QED}\ge0.6$, and $\text{SA}\le4$) got significantly improved.
>
>
> In the case where “high quality” refers to better coverage of chemical space, we did not directly optimize Eq. 9 (previously Eq. 3 before revision) because it is expensive to optimize computationally. Instead, we incorporated several chemical space measures (e.g., Diversity, SumBottleneck, and #Circles) in MARS (Eq. 11-13 in Appendix E) to encourage chemical space exploration as an approximation for Eq. 9 . Through experiments, we demonstrated that this simple approximation can already improve $\mu(\mathcal{S})$ to a great extent as shown in Table 2. We believe that a better optimization for Eq. 9 is much needed and this can be a research direction that is worth exploring in the future.
>
> > **Q4: The model equipped with this metric did not show significant improvement over other metrics.**
>
> We would like to clarify that the models equipped with chemical space measures (i.e., MARS variants) have shown significant improvements over many metrics. Table 2 and Table 5 listed MARS variants’ performances on multiple chemical space measures. Even though MARS+#Circles only explicitly encouraged improving #Circles during optimization, the performances on Diversity (by 0.013), Sumbottleneck (by 1.39 times), and #FG (by 1.74 times) were all significantly improved. The same applies to MARS+Diversity (#Circles (0.75) by 1.8, Sumbottleneck by 1.34 times, #FG by 1.34 times) and MARS+Sumbottleneck (#Circles (0.75) by 6.7, Diversity by 0.016, #FG by 2.51 times). We also observed that #RS (number of ring systems) and #BM (number of Bemis-Murcko scaffolds) were not improved significantly, and this might be due to the fact that the upper limits of #RS and #BM are correlated with the overall molecular weights. During the generation of hit-like molecules, these two metrics could have an early convergence, since they are both constrained by the number of possible combinations of the molecular core structures/backbones.

---

> ### Author Response · Authors · 2022-11-18
> **Thank the reviewer for the constructive comments! Hope our response addresses the concerns.**
>
> Dear Reviewer yinU,
>
> As the end of the response period is approaching, we would like to gently follow up and see if our response addresses your concerns. If there are any further concerns, we really appreciate it if you could kindly let us know, and we will try our best to address them in the remaining of the response period. Thank you again for your time and constructive comments!
>
> Authors

---

> > ### Comment · Reviewer_yinU · 2022-11-25
> > **Thanks for the response**
> >
> > Thanks for the detailed response and revision that have addressed my concerns. I have already raised the score.

---

### Author Response · Authors · 2022-11-17
**Response to all the reviewers**

We would like to express our sincere appreciation to all reviewers for your encouraging and constructive feedback on our work. Below we summarize some common questions raised by reviewers and provide our response to them. We have also replied to each reviewer with a more detailed point-to-point response.

> **Q1: Axiom 4.1 (Subadditivity) and Axiom 4.2 (Dissimilarity).**

For **Axiom 4.1 (Subadditivity)**, we have now elaborated more on our motivation for introducing the constraint for subadditivity.


For **Axiom 4.2 (Dissimilarity)**, following the suggestions from reviewers, we have now reformulated this axiom in the revised submission so that it more directly matches our description and motivation. Please refer to Sec. 4.1 for more details.

We also re-examined the dissimilarity property for all example measures under the new axiom (see Appendix B). The results are mostly consistent except that the SumDiameter measure can satisfy the dissimilarity constraint now, although it still doesn’t satisfy the subadditivity constraint (Fig. 1).

We would like to stress that even though we have reformulated one of our axioms, *the main conclusions and contributions of this paper remain unchanged*:

1. Researchers can still follow the proposed evaluation framework to compare potential chemical space measures theoretically and empirically;
2. The #Circles still stands out among all the considered measures as the only measure that satisfies both subadditivity *and* dissimilarity, and it performs extraordinarily in the empirical studies;
3. The results of measuring chemical space covered by databases and machine-generated molecules remain unchanged.

> **Q2: The dependence (or sensitivity) of the distance threshold $t$ for the #Circles measure.**

First, we want to clarify that we actually have included a sensitivity test of the threshold $t$ in the original submission (Appendix C.2 and C.3). From the results (Fig. 8 and Fig. 11), we can see that the variation of #Circles with respect to $t$ is not sharp in the range of $[0.40, 0.75]$ in the fixed-size setting and $[0.40, 0.80]$ in the growing-size setting. Both settings agree that the optimal value of $t$ is around $0.75$.

In addition, we want to stress that the parameter $t$ has a meaningful interpretation. When $t$ is smaller, the corresponding #Circles measure tends to reflect the "spread" of molecules at a finer granularity. When $t$ is larger, the corresponding #Circles focuses more on the global layout of the molecule distribution. In the revised version, we also added further investigation on how #Circles varies according to $t$ in the molecular generation setting. The #Circles curves are displayed in Appendix E.6 (Fig. 19), which reflects the characteristics of various molecular generation methods.

> **Q3: Code implementation and the fast approximation algorithm for #Circles.**

We thank all the reviewers for your interest in our code implementation. We have uploaded our code to an anonymous site. For the implementation of chemical space measures and the fast approximation algorithms of #Circles, please refer to the repository https://anonymous.4open.science/r/chem-measure-1357. For the implementation of MARS and its variants obtained by incorporating chemical space measures, please refer to the repository https://anonymous.4open.science/r/MARS-explore-B4D2/.

Thanks again for your valuable time reviewing our submission and the response! We hope our revision has addressed your concerns. Below please find our detailed response to each reviewer.

---

### Author Response · Authors · 2022-11-17
**Changes made in the revised submission**

Specifically, we have made the following changes in our revised submission (highlighted in blue in the paper):

1. We have renumbered all the equations in the paper (note some equation labels may become different from the labels in the original submission).
1. We have fixed the definition of the Tanimoto distance in Eq. 1.
2. We have slightly changed the categorization system of chemical space measures. Now the *distance-based* measures are renamed as *aggregation-based* measures for clarification.
3. We have fixed the definition of #Circles in Eq. 4 by imposing that $x\ne y$.
4. We have further elaborated our motivations for Axiom 4.1 (Subadditivity) in Sec. 4.1.
5. We have reformulated Axiom 4.2 (Dissimilarity) by following the suggestions from reviewers. After re-proving the dissimilarity property for all example measures (Appendix B), we found the SumDiameter measure satisfies this requirement now. We also have changed Fig. 1 accordingly in the revision.
7. We have clarified our evaluation setup for comparing filtered databases and generation models in Sec. 5.2.2. We highlighted that both the molecules obtained from databases and the molecules generated by models have been examined with the property filtering rules (*i.e.*, $\text{JNK3}\ge0.5$, $\text{QED}\ge0.6$, and $\text{SA}\le4$).
8. We have elaborated more on the potential applications of our proposed measures as well as the generic evaluation framework in other ML and scientific discovery domains in the Conclusion section.
9. We have discussed the theoretical property of a new reference-based measure suggested by R2 at the end of Appendix B.
10. We have updated Fig. 11 in Appendix C.3 where we tested more $t$ values for the growing-size setting.
11. We have presented the molecular property distributions over the candidates generated by MARS and its variants in Appendix E.4 (Fig. 14-15).
12. We have moved the dynamics of #Circles during the generation process of MARS to Appendix E.5 (Fig. 18) due to page limitation.
13. We have presented an intuitive way of characterizing different molecular generation methods by examining the #Circles values with different thresholds $t$ in Appendix E.6 (Fig. 19).
14. We have described the fast approximation algorithms for computing #Circles in Appendix F (Algorithm 3-4).
15. We have fixed typos as suggested by the reviewers.

---

### Comment · Reviewer_UhAH · 2023-02-25
**Suggestion to discuss relationship between #Circles and packing number**

In a recent discussion of this paper, a colleague of mine pointed out that the `#circles` metric described in this paper is essentially the same as the _packing number_ in topology. [Wikipedia](https://en.wikipedia.org/wiki/Covering_number) defines the packing number with radius $r$ as equivalent to #circles at $2r$, while [these lecture notes](http://www.stat.yale.edu/~yw562/teaching/598/lec14.pdf) define it as being equivalent to #circles. Overall I am pretty sure that the definition of #circles is _exactly_ equivalent up to a change of variables, but the authors should double check this. At the very least, it is extremely similar. My guess is that the authors of the paper were not aware of these connections.

I feel slightly embarrassed that I did not catch this during my original review. I think it detracts slightly from the paper (because #circles is not actually a _novel_ metric), but nonetheless _I still think that accepting this paper was the correct decision_: the idea to use it as a diversity measure for molecule generation is a valuable insight, and the analysis of other diversity measures is valuable.That being said, in light of this, I think the following changes would be appropriate:

1. The authors should acknowledge that #circles and packing number are equivalent (or at least very similar), and add some sort of citation to refer a reader to relevant work in this area (perhaps a topology textbook of some sort). I also think the authors should not say that their metric is "novel".
2. Previous work has proved many theorems about packing number, for example that it is non-decreasing, and have also characterized the relationship with the _covering number_. Perhaps these studies could be leveraged to strengthen the claims and conclusions of the paper?

---

> ### Author Response · Authors · 2023-03-04
> **Response to the suggestion on discussing the relationship between #Circles and the packing number**
>
> Thank you very much for pointing out this connection! In the camera-ready version, we have indicated this connection in our main paper, and we clarified the connection in detail in the appendix (Appendix B). We also made sure that we claim the proposed metric as a *"new measure of the chemical space coverage"* instead of a "novel metric" in general.
>
> To our best knowledge, the application of this packing measure in chemistry is new. Indeed, leveraging the insights from topology literature to strengthen the analysis of the chemical space #Circles metric is a very interesting future direction, which we also pointed out in the camera-ready version. Thanks again for pointing out this connection, and we have expressed our gratitude in the acknowledgment session.

---

### Decision · Program_Chairs · 2023-01-20

**Decision:**

Accept: poster

**Justification For Why Not Higher Score:**

The paper is acceptable, but still with some concerns from the reviewers.

**Justification For Why Not Lower Score:**

The paper has its merits, and after the discussions, one of the reviewers raised his/her score to reflect the consensus.

**Metareview: Summary, Strengths And Weaknesses:**

The paper proposes a new metric called #Circles for chemical spacing measurement. The authors analyzed and compared existing metrics and their metric mathematically, proving that their metric has two good properties. They also conducted experiments to analyze current datasets and molecule generation models by their metric.

The paper originally obtained diverse reviews. By considering the author rebuttals and after several rounds of discussions, the reviewers eventually come to a consensus and believe the paper has its merits and is worth accepting. In particular,
1)	It is working on an important topic – the quality of the metric of ML-based de novo drug design.
2)	The authors have significantly improved the motivation of their work during their revisions.
3)	The paper should have implications for the ICLR audience.

**Note From Pc:**

if the above contains the word "oral" or "spotlight" please see: "oral" presentation means -> notable-top-5% and "spotlight" means -> notable-top-25%. As stated in our emails, we are disassociating presentation type from AC recommendations

**Summary Of Ac-Reviewer Meeting:**

We had an in-depth discussion among the reviewers. Below are some notes:

UhAH: "I am leaning towards acceptance but still have some reservations, which is why I gave a score of 6. For me, the biggest pro of this paper is that it points out a deficiency in existing, commonly-used diversity measures in ML, and proposes a replacement which I think is strictly superior. The biggest con is the writing: up until a recent draft in the rebuttal, the key claims of the paper relied on their axioms being "simple and intuitive" to the reader, which I think they were not. I think the authors have improved this greatly during the rebuttal period by providing more substantive arguments, but I think it could be improved further. Overall, the pros outweigh the cons here which is why I gave a score of 6. I would rather see this paper accepted than rejected, but would also like to see the authors improve the arguments in their paper for the camera-ready version."

cGEX: "I am suggesting to accept this paper because (1) This paper is working on a very important problem (as all of the reviewers agree to some extent), and (2) This paper has no fatal error (though there are several concerns), meaning that, there exists a trade-off between pros and cons, and I consider the cons are not very significant as compared to the pros. In addition, although I had a concern about the axioms, they have been improved in the revised manuscript, and now are easier to understand. Therefore, I suggest to see this paper get accepted."

74LY: "In my view, this was a strong paper to begin with - it addresses an often forgotten aspect in ML-based de novo drug design: that of the quality of its metrics. As some of the other reviewers, my initial reservations were in regards to the claims that some of the proposed axioms were "intuitive". In my opinion, the authors have improved on motivation for these during the revision process and their work should be accepted. "

yinU: "After reading the author's response and other reviewers' comments, I have already raised the score."